# ANALYZING ATTENTION MECHANISMS THROUGH LENS OF SAMPLE COMPLEXITY AND LOSS LANDSCAPE

## ABSTRACT

Attention mechanisms have advanced state-of-the-art deep learning models for many machine learning tasks. Despite significant empirical gains, there is a lack of theoretical analyses on their effectiveness. In this paper, we address this problem by studying the sample complexity and loss landscape of attention-based neural networks. Our results show that, under mild assumptions, every local minimum of the attention model has low prediction error, and attention models require lower sample complexity than models without attention. Besides revealing why popular self-attention works, our theoretical results also provide guidelines for designing future attention models. Experiments on various datasets validate our theoretical findings.

## 1 INTRODUCTION

Significant research in machine learning has focused on designing network architectures for superior performance, faster convergence and better generalization. Attention mechanisms are one such design choice that is widely used in many natural language processing and computer vision tasks. Inspired by human cognition, attention mechanisms advocate focusing on relevant regions of input data to solve a desired task rather than ingesting the entire input.

Several variants of attention mechanisms have been proposed, and they have advanced the state of the art in machine translation (Bahdanau et al., 2014; Luong et al., 2015; Vaswani et al., 2017), image captioning (Xu et al., 2015), video captioning (Pu et al., 2018), visual question answering (Zhou et al., 2015; Lu et al., 2016), generative modeling (Zhang et al., 2018), etc. In computer vision, spatial/spatiotemporal attention masks are employed to focus only on relevant regions of images/video frames for underlying downstream tasks (Mnih et al., 2014). In natural language tasks, where input-output pairs are sequential data, attention mechanisms focus on the most relevant elements in the input sequence to predict each symbol of the output sequence. Hidden state representations of a recurrent neural network are typically used to compute these attention masks.

Substantial empirical evidence on the effectiveness of attention mechanisms motivates us to study the problem from a theoretical lens. To this end, it is important to understand the loss landscape and optimization of neural networks with attention. Analyzing the loss landscape of neural networks is an active ongoing research area, and it can be challenging even for two-layer neural networks (Poggio & Liao, 2017; Rister & Rubin, 2017; Soudry & Hoffer, 2018; Zhou & Feng, 2017; Mei et al., 2018b; Soltanolkotabi et al., 2017; Ge et al., 2017; Nguyen & Hein, 2017a; Arora et al., 2018). Convergence of gradient descent for two-layer neural networks has been studied in Allen-Zhu et al. (2019); Mei et al. (2018b); Du et al. (2019). Ge et al. (2017) shows that there is no bad local minima for two-layer neural nets under a specific loss landscape design. These works reveal the importance of understanding loss landscape of neural networks.

Unfortunately, these results cannot be directly applied for attention mechanisms. In attention models, the network structure is different, and the attention introduces additional parameters that are jointly optimized. To the best of our knowledge, there is no existing work analyzing the loss landscape and optimization of attention models. In this work, we present theoretical analysis of self-attention models (Vaswani et al., 2017), which uses correlations among elements of input sequence to learn an attention mask.

We summarize our work as follows. We carefully analyze attention mechanisms on the loss landscape in Sections 3 and 4. In Section 3, we show that, under mild assumptions, every stationary point of attention models achieves a low generalization error. Section 4 studies other properties of attention models on the loss landscapes. After the loss landscape analyses, we discuss how our theoretical results can guide the practitioners to design better attention models in Section 5. Then we validate our theoretical findings with experiments on various datasets in Section 6. Section 7 includes a few concluding remarks. Proofs and more technical details are presented in the appendix.

## 2 ATTENTION MODELS

Attention mechanisms are modules that help neural networks focus only on relevant regions of input data to make predictions. To compare attention model with non-attention model, we first introduce a two-layer non-attention model as the baseline model. The network architecture consists of a linear layer followed by rectified linear units (ReLU) as a non-linear activation function, and a second linear layer. Denote the weights of the first layer by $\boldsymbol{w}^{(1)} \in \mathbb{R}^{p \times d}$, the weights of the second layer by $\boldsymbol{w}^{(2)} \in \mathbb{R}^d$, and the ReLU function by $\phi(\cdot)$. Then the response function for the input $\boldsymbol{x} \in \mathbb{R}^p$ can be written as $y = \boldsymbol{w}^{(2)T} \phi(\langle \boldsymbol{w}^{(1)}, \boldsymbol{x} \rangle)$. We call the above function "baseline model" since it does not employ any attention.

To study such mechanisms, we mainly focus on analyzing the most popular self-attention model. In this paper, we consider two types of self-attention model.

For the first type of self-attention model, we consider attention weights that are determined by a function $\boldsymbol{f}(\boldsymbol{x})$: $y = \boldsymbol{w}^{(2)T} \phi(\langle \boldsymbol{w}^{(1)}, \boldsymbol{x} \odot \boldsymbol{f}(\boldsymbol{x}) \rangle)$ where $\boldsymbol{f}(\cdot)$ is a known mapping function from $\mathbb{R}^p$ to $\mathbb{R}^p$, representing the attention weight of each feature with any given $\boldsymbol{x}$. This model is a prototype version of transformer model (Vaswani et al., 2017), with a pre-determined function as attention weights.

Second, we introduce a more practical self-attention setup, which is the transformer model proposed in Vaswani et al. (2017). To mimic the NLP task, we set the input $\boldsymbol{x}_i = (\boldsymbol{x}_i^1, \ldots, \boldsymbol{x}_i^p) \in \mathbb{R}^{t \times p}$, where $\boldsymbol{x}_i^j \in \mathbb{R}^t$, are t-dimensional vectors. Intuitively, each $\boldsymbol{x}_i$ corresponds to independent sentences for $i = 1, \ldots, n$, and $\boldsymbol{x}_i^j$'s are fixed dimensional vector embedding of each word in sentence $\boldsymbol{x}_i$. $\boldsymbol{w}^Q, \boldsymbol{w}^K \in \mathbb{R}^{d_q \times t}$ are query and key weight matrices, and $\boldsymbol{w}^V \in \mathbb{R}^{d_v \times t}$ is the value matrix. For each input $\boldsymbol{x}_i$, the key is calculated as: $\boldsymbol{K}_i = (\boldsymbol{w}^K \boldsymbol{x}_i)^T \in \mathbb{R}^{p \times d_q}$; For $z^{th}$ vector in the input, the query vector is computed as: $\boldsymbol{Q}_i^z = (\boldsymbol{w}^Q \boldsymbol{x}_i^z)^T \in \mathbb{R}^{1 \times d_q}$ for $z = 1, \ldots, p$. The value matrix $\boldsymbol{V} = \boldsymbol{w}^V \boldsymbol{x}_i \in \mathbb{R}^{d_v \times p}$. Then the self-attention w.r.t to the $z^{th}$ vector in the input $\boldsymbol{x}_i$ is computed as:

$$\boldsymbol{a}_i^{self(z)}(\boldsymbol{x}_i^z, \boldsymbol{w}^Q, \boldsymbol{w}^K) = softmax(\frac{\boldsymbol{Q}_i^z \boldsymbol{K}_i^T}{\sqrt{d_q}}) \tag{1}$$

for $z = 1, \ldots, p$. And $\boldsymbol{a}_i^{self} = (\boldsymbol{a}_i^{self(1)}, \ldots, \boldsymbol{a}_i^{self(p)})$. This self-attention vector represents the interaction between different words in each sentence. The value vector for each word in the sentence $\boldsymbol{x}_i^z$ can be calculated as $\boldsymbol{V}_i^z = \boldsymbol{V} \boldsymbol{a}_i^{self(z)} \in \mathbb{R}^{d_v}$. This value vector is then passed to a 2-layer MLP parameterized by $\boldsymbol{w}^{(1)} \in \mathbb{R}^{p d_v \times d}$ and $\boldsymbol{w}^{(2)} \in \mathbb{R}^{d \times 1}$, resulting in the following general model:

$$y_i = \boldsymbol{w}^{(2)T} \phi(\langle \boldsymbol{w}^{(1)}, vec(\boldsymbol{w}^V \boldsymbol{x}_i \boldsymbol{a}_i^{self}) \rangle) + \epsilon_i \tag{2}$$

where $vec(\cdot)$ represents the vectorization of a matrix, and $\epsilon_i$ are i.i.d sub-Gaussian error.

## 3 SAMPLE COMPLEXITY ANALYSES

In this section, we focus on analyzing the loss landscape for the the self-attention model as introduced in Section 2. In Section 3.1, we consider the sample complexity of the model with known attention weight function $\boldsymbol{f}(\boldsymbol{x})$. In Section 3.2, we consider transformer self-attention model, in which the attention weight function is also need to be learnt. Section 3.3 discusses the sample complexity result for multi-layer self-attention model.

To avoid the non-differentiable point of ReLU $\phi$, we use the softplus activation function $\phi_{\tau_0}(x)$, i.e., $\phi_{\tau_0}(x) = \frac{1}{\tau_0} \log(1 + e^{\tau_0 x})$ Note $\phi_{\tau_0}$ converges to ReLU as $\tau \to \infty$ (Glorot et al., 2011). All

theoretical results we derived with $\phi_{\tau_0}(x)$ holds for any arbitrarily large $\tau_0$. For ease of notation, we still use $\phi(x)$ to denote the softplus function.

In this paper, we focus on the regression task which minimizes the following loss function:$L = \mathbb{E}_{(\boldsymbol{x},y)\sim D}\|h(\boldsymbol{x}) - y\|_2^2$, where $h(\boldsymbol{x})$ is the defined baseline/attention models. Our theory can be easily extended to classification tasks as well.

## 3.1 Asymptotic property of self-attention model with known weight function

We start with a self-attention model with known attention function $\boldsymbol{f}(\cdot)$. The objective function is:

$$\min_{\boldsymbol{w}}\frac{1}{2n}\sum_{i=1}^{n}(\boldsymbol{w}^{(2)T}\phi(\langle\boldsymbol{w}^{(1)}, \boldsymbol{x}_i \odot \boldsymbol{f}(\boldsymbol{x}_i)\rangle) - y_i)^2 \tag{3}$$

where $\|f(\boldsymbol{x})\|_2 = 1$, as the sum one attention weights.

Before proceeding, we introduce three major assumptions for our analysis: (A1): Model output $y$ can be specified by a two-layer neural network with attention model structure; (A2): The Attention weights are mainly contributed by top several masks; (A3): Hidden layers of the network are sufficiently overparameterized such that they have sufficient expressiveness to predict $y$.

For space consideration, we provide the detailed justification of these assumptions and explain why they can be intuitively summarized as (A1)-(A3) in the Appendix (A.1). Specifically, (A1) is supported by previous works. For (A2) and (A3), we verify them both theoretically and empirically in Appendix Section A.1 and in our experiments Section 6. In short, (A2) naturally holds since softmax function in attention model helps us to obtain a concentrated attention weights instead of evenly spread out weights; (A3) naturally holds for overparameterized neural networks. More details can be found in Appendix Section A.1. In the following, we present their mathematical presentations.

Let $\phi(\langle\boldsymbol{w}^{(1)}, \boldsymbol{x} \odot \boldsymbol{f}(\boldsymbol{x})\rangle)$ represents the $d$-dimensional random vector of hidden units with network weights $\boldsymbol{w}^{(1)}$. For any vector $\boldsymbol{z}_1$ and $\boldsymbol{z}_2$, we denote $var(\boldsymbol{z}_1|\boldsymbol{z}_2)$ as the covariance matrix of residual $\boldsymbol{z}_1$ after taking linear regression on $\boldsymbol{z}_2$. The explicit form can be represented as $var(\boldsymbol{z}_1|\boldsymbol{z}_2) = \Sigma_{z_1z_1} - \Sigma_{z_2z_1}\Sigma_{z_1z_1}^{-1}\Sigma_{z_1z_2}$, where $\Sigma_{z_1z_2}$ represents the cross-covariance matrix of vector $\boldsymbol{z}_1$ and $\boldsymbol{z}_2$. The explicit derivation can be found on page 176 of Izenman (2013). In our setting, $\boldsymbol{z}_1 = \phi(\langle\boldsymbol{w}^{(1)\star}, \boldsymbol{x} \odot \boldsymbol{f}(\boldsymbol{x})\rangle$, and $\boldsymbol{z}_2 = \phi(\langle\boldsymbol{w}^{(1)}, \boldsymbol{x} \odot \boldsymbol{f}(\boldsymbol{x})\rangle$.

(A1) There exists a set of parameters $(\boldsymbol{w}^{(1)\star}, \boldsymbol{w}^{(2)\star})$ such that $y_i = \boldsymbol{w}^{(2)\star T}\phi(\langle\boldsymbol{w}^{(1)\star}, \boldsymbol{x}_i \odot \boldsymbol{f}(\boldsymbol{x})\rangle) + \epsilon_i$ for $i = 1, 2, ...n$, where $\epsilon_i$'s follow the sub-Gaussian distribution $subG(0, C_3^2)$ and $\boldsymbol{x}_i \perp\!\!\!\perp \epsilon_i$.

(A2) For any $\boldsymbol{x}_i$, we order the attention weights $\boldsymbol{f}(\boldsymbol{x}_i)$ in descending order as $\boldsymbol{f}(\boldsymbol{x}_i)_{(1)}, \ldots, \boldsymbol{f}(\boldsymbol{x}_i)_{(p)}$. There exist positive integer $0 < s_0 < p$ and $0 < \tau < 1$, such that largest $s_0$ attention weights $\boldsymbol{f}_{s_0} = \{\boldsymbol{f}(\boldsymbol{x}_i)_{(1)}, \ldots, \boldsymbol{f}(\boldsymbol{x}_i)_{s_0}\}$ satisfies $\|\boldsymbol{f}_{s_0}\|_2 \geq 1 - \tau$.

(A3) For $\gamma > 0$, $\lambda_{\max}(Var(\phi(\langle\boldsymbol{w}^{(1)\star}, \boldsymbol{x}_i \odot \boldsymbol{f}(\boldsymbol{x})\rangle)|\phi(\langle\boldsymbol{w}^{(1)}, \boldsymbol{x} \odot \boldsymbol{f}(\boldsymbol{x})\rangle)) = o(\gamma^2)$ as $n \gtrsim \frac{k^4}{\gamma^2}\log(\frac{k}{\gamma})(pd + d)$ and $n \to \infty$ for a constant k which satisfies $k < \sqrt{s_0} + \tau\sqrt{p}$.

Other than these three major assumptions, we also assume the regularity assumption (B1): the model samples $\boldsymbol{x}_i$'s are i.i.d. with maximum bound; network weights $\boldsymbol{w}^{(1)}$ and $\boldsymbol{w}^{(2)}$ have bounded $\ell_2$ norms, denoted by $C_1$ and $C_2$, respectively; and the output $y$ is centralized. The regularity assumption (B1) is standard in the literature, which will be justified in the Appendix (A.2). Given these assumptions, we show that the sample complexity bound of attention model:

**Theorem 1.** *(Sample complexity for attention model)* Under (A1)–(A3) and regularity assumption (B1), for any $\gamma > 0$ and $s_0$ s.t. $k \lesssim \sqrt{s_0} + \tau\sqrt{p}$, given the sample size

$$n \gtrsim \frac{(\sqrt{s_0} + \tau\sqrt{p})^4\eta^2}{\gamma^2}\log(\frac{(\sqrt{s_0} + \tau\sqrt{p})\eta}{\gamma})(pd + d)$$

where $\eta = C_1^2C_2$. with probability converging to 1, any stationary point $(\tilde{\boldsymbol{w}}^{(1)}, \tilde{\boldsymbol{w}}^{(2)})$ of the objective function (3) satisfies that: $\mathbb{E}(\tilde{\boldsymbol{w}}^{(2)T}\phi(\langle\tilde{\boldsymbol{w}}^{(1)}, \boldsymbol{x} \odot \boldsymbol{f}(\boldsymbol{x})\rangle) - \mathbb{E}(y|\boldsymbol{x}))^2 \lesssim \gamma^2$.

The proof sketch and complete proof are provided in Appendix Section E. To explicitly compare the sample complexity bound of our attention model with non-attention model, we provide the following corollary for sample complexity bound of baseline model. We just need to set $s_0 = 0$ and $\tau = 1$ in Theorem 1, fixing all attention weights to be equal, then the model is the baseline one as follows.

**Corollary 1.** *(Sample complexity for non-attention model) Under (A1)–(A3) and regularity assumption (B1), for non-attention model with $f(x_j) = 1/\sqrt{p}$ for all features $x_j \in \boldsymbol{x}$, given the sample size:*

$$n \gtrsim \frac{p^2\eta^2}{\gamma^2}\log(\frac{p\eta}{\gamma})(pd+d)$$

*where $\eta = C_1^2 C_2$. With probability converging to 1, any stationary point $(\tilde{\boldsymbol{w}}^{(1)}, \tilde{\boldsymbol{w}}^{(2)})$ of the objective function (3) satisfies that: $\mathbb{E}(\tilde{\boldsymbol{w}}^{(2)^T}\phi(\langle \tilde{\boldsymbol{w}}^{(1)}, \boldsymbol{x} \odot \boldsymbol{f}(\boldsymbol{x})\rangle)) - \mathbb{E}(y|\boldsymbol{x}))^2 \lesssim \gamma^2$.*

There is an important message from the comparison of Theorem 1 and Corollary 1. The corresponding sample complexity bound from Corollary is in higher order than the sample complexity in Theorem 1. This fact can be observed as follows. We can see the effect of concentration in attention models from Theorem 1. In Theorem 1, the sample complexity bound is proportional to $(\sqrt{s_0} + \tau\sqrt{p})^4$. When the attention is sufficiently concentrated, it means that we have a much lower order of $s_0 \ll p$ and $\tau \to 0$, then our sample complexity will be significantly reduced.

Also from the sample complexity bound of Theorem 1, up to a log term, prediction error $\gamma$ is proportional to $n^{-1/2}$, which is the optimal rate of convergence in regression. In Imaizumi & Fukumizu (2018) and similar works, they showed the generalization error convergence rate is $O(n^{-t})$ where $0 < t < 1/2$. These facts all imply that the bound is tight in aspect of sample size compared with existing works.

## 3.2 ASYMPTOTIC PROPERTY OF SELF-ATTENTION MODEL WITH UNKNOWN WEIGHT FUNCTION

In this section, we extend our analyses to the transformer self-attention model introduced in Section 2.

Theorem 1 implies that if the self-attention mask can be precisely computed with $\boldsymbol{f}(\boldsymbol{x})$, we can derive its sample complexity bound. However, the function $\boldsymbol{f}(\cdot)$ is not necessarily known, and needs to be learned in real-world applications. Therefore, the transformer self-attention setup is more desired in real-world settings. It provides a concrete model to learn the parameter of attention weight function. Denoting $\boldsymbol{w} = (\boldsymbol{w}^{(1)}, \boldsymbol{w}^{(2)}, \boldsymbol{w}^Q, \boldsymbol{w}^K, \boldsymbol{w}^V)$, the two-layer self-attention model can be formulated as:

$$\min_{\boldsymbol{w}}\frac{1}{2n}\sum_{i=1}^{n}(\boldsymbol{w}^{(2)T}\phi(\langle \boldsymbol{w}^{(1)}, vec(\boldsymbol{w}^V \boldsymbol{x}_i \boldsymbol{a}_i^{self})\rangle)) - y_i)^2 \tag{4}$$

We now introduce necessary assumptions.

(A4) For each column $\boldsymbol{a}_i^{self}$, we order the entries of $\boldsymbol{a}_i^{self}$ in descending order as $\boldsymbol{a}_{i(1)}^{self}, \ldots, \boldsymbol{a}_{i(p)}^{self}$. There exist positive integer $0 < s_0 < p$ and $0 < \tau < 1$, such that the largest $s_0$ attention weights $\boldsymbol{a}_{i(leading)}^{self} = \{\boldsymbol{a}_{i(1)}^{self}, \ldots, \boldsymbol{a}_{i(s_0)}^{self}\}$ satisfies $\|\boldsymbol{a}_{i(leading)}^{self}\|_2 \geq 1 - \tau$.

(A5) Denote $\boldsymbol{\Phi}^\star = \phi(\langle \boldsymbol{w}^{(1)\star}, vec(\boldsymbol{w}^{V\star}\boldsymbol{x}_i \boldsymbol{a}_i^{self\star})\rangle)$, and $\boldsymbol{\Phi} = \phi(\langle \boldsymbol{w}^{(1)}, vec(\boldsymbol{w}^V \boldsymbol{x}_i \boldsymbol{a}_i^{self})\rangle)$. For any $\gamma > 0$ we have $\lambda_{\max}(Var(\boldsymbol{\Phi}^\star|\boldsymbol{\Phi})) = o(\gamma^2)$. as $n \gtrsim \frac{k^2 d_v d}{\gamma^2}\log(\frac{(k+\sqrt{p}t)}{\gamma})$ and $n \to \infty$.

where similar to (A1), $(\boldsymbol{w}^{(1)\star}, \boldsymbol{w}^{(2)\star}, \boldsymbol{w}^{V\star}, \boldsymbol{a}^{self\star})$ correspond to the true parameter set such that

$$y_i = \boldsymbol{w}^{(2)\star T}\phi(\langle \boldsymbol{w}^{(1)\star}, vec(\boldsymbol{w}^{V\star}\boldsymbol{x}_i \boldsymbol{a}_i^{self\star})\rangle) + \epsilon_i. \tag{5}$$

(A4) and (A5) are parallel to (A2) and (A3), assuming the attention weights are focused on $s_0$ items, and a sufficient expressive power of $\boldsymbol{\Phi}$. Furthermore, sparse transformer Child et al. (2019), sparsemax attention model Martins & Astudillo (2016) and local attention model Luong et al. (2015) can be regraded as a special case of the assumption with $\tau = 0$, in which we only consider attention weights between partial locations instead of all locations. We further verify this assumption in our

experiments that $s_0 \ll p$ and $\tau \to 0$. Other regularity assumptions on feature and parameter space are similar with the ones in Theorem 1. We denote is as (B2): The $\ell_2$ norm bound for $\boldsymbol{w}^{(1)}$, $\boldsymbol{w}^{(2)}$,$\boldsymbol{w}^K$ and $\boldsymbol{w}^Q$ are, respectively, $C_1, C_2, C_5$, and $C_6$, and the $\ell_1$ column norm of $\boldsymbol{w}^V$ is $C_8$. They are presented in Appendix (A.3). Now we provide sample complexity bound for transformer self-attention models:

**Theorem 2.** *(Sample complexity for transformer self-attention model) Under (A4), (A5) and regularity assumptions (B2), for any $\gamma > 0$ and $s_0$ s.t $k \lesssim s_0$, given the sample size:*

$$n \gtrsim \frac{((\sqrt{s_0} + \tau\sqrt{p})^2 + \sqrt{p}t)^2 p d_v d\eta_a^2}{\gamma^2} \log(\frac{((\sqrt{s_0} + \tau\sqrt{p})^2 + \sqrt{p}t)\eta_a\eta_b}{\gamma})$$

*where $\eta_a = C_1^2 C_2 C_8$ and $\eta_b = C_5 + C_6$, with probability tending to 1, any stationary point $(\tilde{\boldsymbol{w}}^{(1)}, \tilde{\boldsymbol{w}}^{(2)}, \tilde{\boldsymbol{w}}^Q, \tilde{\boldsymbol{w}}^K)$ of the objective function (4) satisfies that: $\mathbb{E}(\tilde{\boldsymbol{w}}^{(2)T}\phi(\langle \tilde{\boldsymbol{w}}^{(1)}, vec(\boldsymbol{w}^V \boldsymbol{x}_i \boldsymbol{a}_i^{self})\rangle)) - \mathbb{E}(y|\boldsymbol{x}))^2 \lesssim \gamma^2$.*

There are two important messages from Theorem 2. First, theorem 2 shows that, with the help of self-attention, we can achieve consistent predictions under a more expressive class of models (equation 5), which is analyzed in Yun et al. (2019). Non-attention model does not have consistency for this class of models. To train a non-attention model on data with self-attention structure, more layers and larger network parameter size are required to reduce such bias. Second, we can see the help of concentrating the attention in designing transformer model. Similar to Theorem 1, we see the sample complexity bound is proportional to $(\sqrt{s_0} + \tau\sqrt{p})^4$. Therefore, a properly small $s_0$ and $\tau$ can significantly reduce the sample complexity of self-attention models. Later in our experiment, we show that it is exactly what is happening in the real transformer model. Our theorem also answers why the self-attention design with softmax function can effectively helps us achieve better prediction results. What's more, it also explains the effectiveness of sparse design in attention model. Sparse attention is one special case of our concentration condition with $\tau = 0$. There has been work showing that sparse attention weights can significantly reduce computational cost and improve the performance, and it is verified in sparse transformer,local attention model and sparsemax attention models(Child et al., 2019; Luong et al., 2015; Martins & Astudillo, 2016).

### 3.3 MULTI-LAYER SELF-ATTENTION MODELS AND RECURRENT ATTENTION MODELS

Theorems 1 and 2 can be extended to multi-layer neural nets. Due to page limit, the rigorous definition, notations, assumptions and statements of Theorem 3 are deferred to Appendix Section A.4. Here we provide a plain statement of it as follows.

**Theorem 3.** *Given the overparameterized and regularity condition, for any given generalization error level $gamma$, with high probability, a multi-layer self-attention model can achieve a generalization error smaller than $\gamma$, given the sufficient large sample size.*

Our analyses can be also extended to recurrent attention models, following the recurrent attention setup in Luong et al. (2015). The analysis on recurrent attention model is deferred to Appendix Section B due to page limits.

## 4 NON-LINEARITY, FLATNESS OF MINIMA AND SMALL SAMPLE SIZE

In this section, we further investigate several additional properties on how attention mechanisms improve the landscape of neural networks and keep the nice properties of baseline models in aspects of reducing unnecessary non-linear regions, sharpness of local minimum and it doesn't affect the loss landscape in small sample case.

### 4.1 ON THE NUMBER OF LINEAR REGIONS

We first study how attention mechanisms affect the number of linear regions (Montufar et al., 2014) in a wide two-layer neural network with attention of known attention weight function, when the number of hidden units is larger than the number of non-zero weights in $\boldsymbol{f}(\boldsymbol{x})$. This result shows how the sparsity/concentration of attention weights effects the non-linearity of loss landscape.

**Theorem 4.** *Assume $\|\boldsymbol{f}(\boldsymbol{x})\|_0 = s_0$, which is the sparsity of the attention mask matrix, and the number of units in the hidden layer $n_1 > s_0$. Then the maximal number of linear regions of the function by a two-layer attention model with ReLU activation function, is lower bounded by $\lfloor \frac{n_1}{s_0} \rfloor^{s_0}$.*

In general, we see the bound of attention model is smaller than the one of baseline model. The corresponding plots on bounds can be found in appendix Section C.1. The result implies that, when appropriate attention mechanism is used, the reduction of number of linear regions leads to a simpler landscape, yet the approximation error remains small.

## 4.2 ON FLATNESS/SHARPNESS OF MINIMA

Many recent works, such as Keskar et al. (2016), argue that flatter local minimum tends to generalize well. However, in a recent study, Dinh et al. (2017) observes that by scale transformation, the minima which are observationally equivalent, can be arbitrarily sharp, and the operator norm of a Hessian matrix can also be arbitrarily large. We show that this fact also holds for the self-attention mechanism, if no $\ell_2$ norm bound on parameter $(\tilde{\boldsymbol{w}}^{(1)}, \tilde{\boldsymbol{w}}^{(2)})$ is imposed. Here we introduce the definition of $\epsilon$-flatness as in Hochreiter & Schmidhuber (1997).

**Definition 1.** *Given $\epsilon > 0$, a minimum $\boldsymbol{\theta}$, and loss $L$, $C(L, \boldsymbol{\theta}, \epsilon)$ is the largest connected set containing $\boldsymbol{\theta}$ such that $\forall \boldsymbol{\theta}' \in C(L, \boldsymbol{\theta}, \epsilon)$, $L(\boldsymbol{\theta}') \leq L(\boldsymbol{\theta}) + \epsilon$, and its volume is called the $\epsilon$-flatness.*

In the following Theorem, we analyze the flatness of stationary point for self-attention model.

**Theorem 5.** *Consider the two-layer ReLU neural network with self-attention mechanism as stated in Section 3.2: $y_i = \boldsymbol{w}^{(2)\star T} \phi(\langle \boldsymbol{w}^{(1)\star}, vec(\boldsymbol{w}^V \boldsymbol{x}_i \boldsymbol{a}_i^{self}) \rangle)$, and a minimum $\boldsymbol{\theta} = (\tilde{\boldsymbol{w}}^{(1)}, \tilde{\boldsymbol{w}}^{(2)}, \boldsymbol{w}^V, \boldsymbol{w}^Q, \boldsymbol{w}^K)$ satisfying that $\tilde{\boldsymbol{w}}^i \neq 0$ for $i = (1), (2), V, Q, K$. For any $\epsilon > 0$, $C(L, \boldsymbol{\theta}, \epsilon)$ has an infinite volume, and for any $M > 0$, we can find a stationary point such that the largest eigenvalue of $\nabla^2 L(\boldsymbol{\theta})$ is larger than $M$.*

Theorem 5 indicates that property on flatness of minima is maintained when attention mechanism is applied. Furthermore, $\ell_2$ norm bound helps remove sharp minima which are bad in generalization. It also coincides with our theoretical and empirical result that a flat minimum are expected to generalize better in general (Keskar et al., 2016). We also dicuss the loss landscape of attention model under small sample size. The results are deferred to Appendix Section C.3.

## 5 GUIDANCE ON IMPROVING THE ARCHITECTURE OF ATTENTION MODELS

In this section, we provide insights into future attention model design through our analyses.

**Regularization:** Our analyses suggest proper regularization is helpful in training an attention model. We can see that $\ell_2$ norm bound $C_1, C_2$ play an important role in sample complexity bound. It implies that an $\ell_1$ and $\ell_2$ regularization on network weights $\boldsymbol{w}^{(1)}$ and $\boldsymbol{w}^{(2)}$ are effective in reducing the sample complexity. In Theorem 5, we also find that imposing constraints and regularization on network weights help remove sharp minima and keep flat minima with good generalization.

**Concentration on attention**: From the discussion of Theorem 1, Theorem 2 and Corollary 1, we conclude that a proper concentration design with small $s_0$ and $\tau$ can significantly reduce sample complexities. Our analysis show that the soft-max design concentrate attention on limited number of entries, which help reduce the sample complexity. In different problems, we can further concentrate the attention by adjusting the temperature of the softmax function. The smaller the temperature, more concentrated the attention weights are.

**Overparameterization in query/key weights** From our analyses, we can see that as $d_q$, the dimension of key and query matrices, increases, the sample complexity will not increase significantly. It indicates that we can obtain high expressive power in attention model through overparameterization in query and key matrices to increase expressiveness, without hurting sample complexity.

## 6 EXPERIMENTS

In this section, we validate our theoretical findings empirically. This section is divided into the following parts: (1) Verification of concentration assumption (A2) and (A4) on Portuguese to English translation task. (2) Ablation study on the IMDB reviews dataset, showing the effectiveness of attention, regularization, and attention-concentration on self-attention and recurrent attention models. (3) Experiments on a constructed noisy-MNIST dataset using self-attention models.

**Experiment 1: Concentration of top attentions**

To verify that our key assumptions/observations (A2) and (A4) hold in the real-world task, we investigate the distribution of attention weights in the transformer model on the tasks of translating Portuguese to English as proposed in Vaswani et al. (2017). Then we randomly select the trained attention weights vector for 100000 different words in training samples. For each attention weight vector with sentence length $p_i$, we calculate the $\ell_2$ norm of largest $\lfloor\sqrt{p_i}\rceil$ number of attention weights. We find the top $\lfloor\sqrt{p_i}\rceil$ weights on average contribute to 90.9% of attention weights, very close to 1. The histogram is plotted as follows Figure 1. We see that most of sum weights are close to 1. The result indicates that the largest $\lfloor\sqrt{p_i}\rceil$ attention weights contribute to most of the attention weights. It indicates that our assumption (A2) and (A4) hold with reasonably small $s_0 \ll p$ and $\tau$, such as $s_0 = \lfloor\sqrt{p}\rceil$.

**Experiment 2: Ablation study on self-attention and recurrent attention on the IMDB reviews dataset:**

We consider the problem of sentiment classification on an IMDB reviews dataset (Maas et al., 2011). The task is to classify the sentiment of a sentence as either a positive or negative one. We zero-pad all our sentences to make their length equal to the sentence length 130. For every input word, we train their embeddings with random initialization (of dimension 100) which is then passed to the neural network. Hence, the dimension of input is $130 \times 100$. We consider baseline 2-layer MLP, and then consider adding self-attention, weight regularization and tempered softmax function into the model to verify our theoretical analysis and corresponding guidance in Section 5.

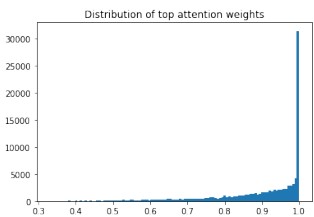

Figure 1: Distribution of $\ell_2$ norm of top $\lfloor\sqrt{p_i}\rceil$ of attention weights

**Baseline model:** To train the baseline model, we first flatten the input one large vector of dimension $130 \times 100$ and pass it to a 1-hidden layer MLP with $h$ hidden units. The model is trained using binary cross entropy loss.

**Self-attention model:** For self-attention model, the dimensions of query, key and value matrices are $\boldsymbol{w}^Q \in \mathbb{R}^{100\times100}$, $\boldsymbol{w}^K \in \mathbb{R}^{100\times100}$, $\boldsymbol{w}^V \in \mathbb{R}^{100\times130}$, respectively. We first compute the attention mask $\boldsymbol{a}_i^{self}$ as per equation 1 ($\boldsymbol{a}_i^{self} \in \mathbb{R}^{130\times130}$). Using this attention mask, the attended feature is then computed as $f_{att} = \boldsymbol{w}^V \boldsymbol{x}_i \boldsymbol{a}_i^{self}$. The feature vector $f_{att} \in \mathbb{R}^{100\times130}$. We then flatten this attended feature and then pass it through a 1-hidden layer MLP with $h$ hidden units.

**Regularization:** We impose a $10^{-4}$ $\ell_1$ regularization on both $\boldsymbol{w}_1$ and $\boldsymbol{w}_2$.

**Tempered Softmax:** We calculate the attention weights as $\boldsymbol{a}_i^{self(z)}(\boldsymbol{x}_i^z, \boldsymbol{w}^Q, \boldsymbol{w}^K) = softmax(\frac{5*\boldsymbol{Q}_i^z \boldsymbol{K}_i^T}{\sqrt{d_q}})$. We multiply the inner product by 5(or temperature as $1/5$). Thus, the softmax operator pushes the small attention weights close to zero while retaining all large attention weights. In this way, it achieves higher level of concentration of attention weights, comparing with the standard softmax function based attention.

**Optimization:** All models were initialized randomly with Xavier initialization. Binary cross-entropy loss was used to train the models. All models were trained using Adam optimizer with a learning rate $10^{-3}$.

To test the sample complexity, we vary the number of training samples in each experiment, train all models and compute the performance on the test set. We varied the fraction of training samples from 1k to 10k of the training data. Each experiment was repeated for 10 replications, and mean and standard deviation was reported in Table 1.

Table 1: Testing accuracy of self-attention ablation study on the IMDB reviews dataset.

| Training sample size | $1k$ | $3k$ | $5k$ | $10k$ |
|---|---|---|---|---|
| Baseline | 0.692 (0.016) | 0.809 (0.016) | 0.821 (0.015) | 0.842 (0.006) |
| Baseline+regularization | 0.737 (0.009) | 0.823 (0.010) | 0.837 (0.005) | 0.841 (0.005) |
| Self Attention | 0.806 (0.007) | 0.821 (0.012) | 0.832(0.007) | 0.861 (0.004) |
| Self Attention+regularization | 0.809 (0.011) | 0.838 (0.011) | 0.853(0.002) | 0.865 (0.006) |
| Self Attention+tempered | 0.798 (0.010) | 0.830 (0.010) | 0.840(0.009) | 0.864 (0.011) |
| Self Attention+regularization+tempered | **0.817 (0.010)** | **0.843 (0.010)** | **0.856 (0.013)** | **0.868 (0.007)** |

**Recurrent attention experiments:** Beyond self-attention, we also provide an ablation study on recurrent attention model, verifying the superiority of recurrent attention model, as our analysis in the Appendix B.

**(1) RNN baseline:** The baseline here is a bi-directional RNN attention model with LSTM cell size 32, and we put the final output of RNN into a 2-hidden layer MLP with hidden units 64 and 20 separately to return the final prediction. **(2) Recurrent attention:** For recurrent attention, we design the structure the same as Luong et al. (2015). **(3) Regularization**: We impose the $\ell_1$ regularization of $10^{-4}$ on network weights in the attention layer. Here we reported the test accuracy results in the following Table 2.

Table 2: Testing accuracy of recurrent attention ablation study on the IMDB reviews dataset.

| Training sample size | $1k$ | $3k$ | $5k$ | $10k$ |
|---|---|---|---|---|
| RNN Baseline | 0.776 (0.023) | 0.804 (0.009) | 0.823 (0.006) | 0.829 (0.015) |
| RNN Baseline+regularization | 0.777 (0.012) | 0.806 (0.010) | 0.814 (0.008) | 0.832 (0.007) |
| Recurrent Attention | 0.789 (0.010) | **0.821** (0.007) | 0.820 (0.009) | 0.835 (0.008) |
| Recurrent Attention+regularization | **0.792** (0.010) | 0.817 (0.009) | **0.834** (0.009) | **0.862** (0.002) |

From the experiment results, we see significant effect of regularization and softmax temperature in self-attention model, and we also see regularization also helps in the recurrent attention model. They all coincide with our theoretical findings.

**Experiment 3: Self-attention on Noisy-MNIST dataset**

To prove the applicability of our analyses, we further verify the effectiveness of attention on image classification task. We construct a noisy MNIST dataset based on the original MNIST dataset. For each original $28 \times 28$ image, we separate a $56 \times 56$ image into 16 square grids, and put the whole digit image randomly into 4 neighboring grids. Finally we generate all other grids with uniform random variables. Examples of our generated dataset are provided in Figure 2.

We consider three models here. **(1) CNN**: We consider a standard 2-layer Convolutional neural networks with fillter size 64, kernel size $3*3$ and max pooling size $2*2$, following a hidden fully-connected layer with size 128. **(2) Self-attention-CNN** We fit a 2-layer convolutional network with filter size 64, kernel size $3 * 3$ with a max pooling size $2 * 2$, obtaining a 1600-dimensional embedding for each of 16 square grids. Then treat these embedding of 16 grids as "16 words embedding" in a sentence, fitting it into equation 2 as our experiment 2. **(3) Equal weight self-attention-CNN**: The model structure is the same as model 2, with fixing the attention weights between all grids are the same. Model (3) is designed as an alternative baseline guaranteeing that the improvement of model 2 is from the attention block instead of the model structure. The testing accuracy of three models are reported in Table 3. Model 1 has 5558k number of parameters, and model 2 has only 670k number of parameters. This fact also guarantees that the superiority of our model is not from training a bigger model.

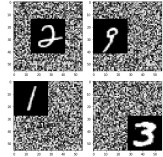

Figure 2: Noisy MNIST dataset

Table 3: Testing accuracy of self-attention on the Noisy MNIST dataset

| Sample size | n=5k | n=20k | n=60k |
|---|---|---|---|
| CNN | 0.874(0.001) | 0.944(0.002) | 0.965(0.002) |
| Self-attention-CNN | **0.970(0.003)** | **0.993(0.000)** | **0.996(0.000)** |
| Equal weight self-attention CNN | 0.900(0.001) | 0.927(0.041) | 0.900(0.029) |

From the table, we see that Attention-CNN model achieves almost perfect testing accuracy as the original MNIST dataset. Although equal-attention CNN has the same expressiveness power, its performance is much worse than attention-CNN model. Again, It shows the usefulness of concentrating self-attention weights properly in the task.

## 7 CONCLUSIONS

In this paper, we study the loss landscape of neural networks on attention models, and show that attention mechanisms help reduce the sample complexity and achieve consistent predictions in the

large sample regime. Besides theoretical analyses of loss landscape, empirical studies validate our theoretical findings. Based on our analyses, we discuss how regularization, concentration on attention, and overparameterization in attention weight matrices can further improve the attention model.

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

APPENDIX

In this appendix, Section A presents detailed theorem assumptions and their justifications; Section B presents the theorem of sample complexity bounds for recurrent attention models; Section C presents details on results in Section 4 of main paper. Section D presents a key lemma of sample complexity bound on attention model with fixed attention masks for all samples; Finally, we provide proofs in section E.

# A    THEOREM ASSUMPTIONS

In this section, we provide more rigorous presentations on the standard assumptions of Theorems 1-3 and further justify these assumptions.

## A.1    JUSTIFICATION OF ASSUMPTIONS

Here we justify (A1) to (A3) for Theorem 1, and explain why they can be intuitively summarized as in the main paper. (A1) assumes the expectation of output $y$ can be specified by a two-layer neural network with attention model structure. It has been studied that general bounded functions with a Fourier representation on $[-1, 1]$ can be well approximated by the defined two-layer network (Barron & Klusowski, 2018). Specific to attention models, Yun et al. (2019) prove the strong expressiveness of transformer type self-attention model give sufficient large network. Therefore it is very mild to assume attention model can identify the mean of output $y$.

(A2) assumes that the attention mask is mostly concentrated on the largest $s_0$ attention weights. This assumption is naturally satisfied due to the softmax function in the self-attention equation. Softmax function makes attention weight proportion to the exponential of the inner product of query and key vectors. This fact significantly enlarges the difference between the inner products, and makes the attention weights do not evenly spread out over all other entries, but only a few of them. Empirically, we verify that this assumption is satisfied in real-world transformer translation task in our experiments. Please check experiment 1 in Section 6 for more details on the distribution of attention weights in Portuguese to English translation. It shows that this assumption is well satisfied.

(A3) assumes that $\{\phi_{1\ldots d}(\langle \boldsymbol{w}^{(1)}, \boldsymbol{x}_i \odot \boldsymbol{f}(\boldsymbol{x})\rangle)\}$ obtain sufficient expressiveness to predict the output $y$ when sample size is sufficiently large. In an overparameterized network with large $d$, we know $\{\phi_{1\ldots d}(\langle \boldsymbol{w}^{(1)}, \boldsymbol{x}_i \odot \boldsymbol{f}(\boldsymbol{x})\rangle)\}$ have up to $2^d$ number of linear regions over $n$ samples. And (A2) assumes these linear regions space all directions of $\phi(\langle \boldsymbol{w}^{(1)\star}, \boldsymbol{x}_i \odot \boldsymbol{f}(\boldsymbol{x})\rangle)$ up to a $o(\gamma)$ term. It says that given the $2^d$ linear regions and strong expressiveness of $\{\phi_{1\ldots d}(\langle w^{(1)}, x \odot f(x)\rangle)\}$, the residual vector is a $o(\gamma)$ term with respect to all directions in $\mathbb{R}^d$. This assumption is parallel to the full column rank condition in (Nguyen & Hein, 2017b), where we essentially assume in overparameterized network, the linear combination spans all directions in $\mathbb{R}^p$. Allen-Zhu et al. (2019) also shows the fact in their Lemma B.1 and Corollary B.2 that in overparameterized networks, in every small region of parameter space, there exists set of parameters with good prediction. And it leads to a good landscape in each neighborhood. These results all indicate that it is reasonable to assume sufficient expressiveness of $\phi(\langle \boldsymbol{w}^{(1)}, \boldsymbol{x} \odot \boldsymbol{a}\rangle)$ in overparameterized networks as $n \to \infty$. What's more, with required large sample size, we can also straightforwardly evaluate this assumption by checking whether these $\phi(\langle \boldsymbol{w}^{(1)}, \boldsymbol{x} \odot \boldsymbol{a}\rangle)$ spread out the whole space and their linear combinations have good estimation on $\boldsymbol{y}$. All these results guarantee that (A3) can be achieved in overparameterized networks. We also empirically validate (A3) by computing the largest eigenvalue of such conditional covariance in (A3). We generate a two-layer network under self-attention model (4) with $n = 500$, $d = 256$, $p = 2142$, $d_v = 100$. We choose the set of parameter same with our experiment setup in Section 6, and hidden layers are sufficient overparameterized with $p = 2142$. Then we compute the empirical conditional covariance $\hat{\lambda}_{\max}(Var(\phi(\langle \boldsymbol{w}^{(1)\star}, \boldsymbol{x}_i \odot \boldsymbol{f}(\boldsymbol{x})\rangle)|\phi(\langle \boldsymbol{w}^{(1)}, \boldsymbol{x} \odot \boldsymbol{f}(\boldsymbol{x})\rangle)) = 3.13 * 10^{-8}$. This result indicates that the largest eigenvalue term $\lambda_{\max}(Var(\phi(\langle \boldsymbol{w}^{(1)\star}, \boldsymbol{x}_i \odot \boldsymbol{f}(\boldsymbol{x})\rangle)|\phi(\langle \boldsymbol{w}^{(1)}, \boldsymbol{x} \odot \boldsymbol{f}(\boldsymbol{x})\rangle))$ is small enough to be assumes as $o(\gamma^2)$ as in (A3).

## A.2 Regularity Assumptions of Theorem 1

In section 3, we described the regularity assumptions of Theorem 1 in words. Here we provide rigorous mathematical representations of these assumptions in the following (A1.1) to (A1.3).

(A1.1) $\boldsymbol{x}_i$ are i.i.d and $\|\boldsymbol{x}_i\|_\infty < C_x$ for any $i = 1, \ldots, n$.

(A1.2) There exist $C_1, C_2, C_\lambda$ such that $\lambda_{\max}(\boldsymbol{\Sigma}_\phi) \leq C_\lambda$, $\|\boldsymbol{w}^{(1)}\|_F < C_1$ and $\|\boldsymbol{w}^{(2)}\|_2 < C_2$ for any $\boldsymbol{w}^{(1)}, \boldsymbol{w}^{(2)} \in \mathcal{S}$.

(A1.3) $\mathbb{E}(\boldsymbol{w}^{(2)}\phi(\langle \boldsymbol{w}^{(1)}, \boldsymbol{x} \odot \boldsymbol{f}(\boldsymbol{x})\rangle)) = \mathbb{E}(y) = 0$.

(A1.1) to (A1.3) are standard assumptions for parameter and feature space. (A1.1) and (A1.2) require upper bounds on the input $\boldsymbol{x}_i$ and $\ell_2$ bound for network weights. It is a standard assumption in landscape analysis (Mei et al., 2018a;b), and also it is crucial to remove sharp minima which may not generalize well (Keskar et al., 2016; Dinh et al., 2017).(See discussions after Theorem 4). These assumptions can be achieved through regularization.

(A1.3) can always be achieved by making the data centered.

## A.3 Assumptions of Theorem 2

Similarly, for transformer-type self-attention, we also provide similar assumptions (A2.1) and (A2.2) as follows:

(A2.1): (A1.2) holds with $\boldsymbol{\Sigma}_\phi = cov(\phi(\langle \boldsymbol{w}^{(1)\star}, vec(\boldsymbol{w}^V \boldsymbol{x}_i \boldsymbol{a}_i^{self(z)}))$. Further there exist $C_5, C_6$ and $C_7$ such that $\|\boldsymbol{w}^Q\|_F \leq C_5$, $\|\boldsymbol{w}^K\|_F \leq C_6$, $\|\boldsymbol{w}^V\|_F \leq C_7$, and $\|\boldsymbol{w}^V\|_1 = \max_{i=1,\ldots,d_v} \sum_{j=1}^p \boldsymbol{w}_{ij}^V \leq C_8$. And $\|\boldsymbol{Q}_i^z \boldsymbol{K}_i^T\|_2 \geq C_9$ for $i = 1, \ldots, n$.

(A2.2) There exists a set of parameters $(\boldsymbol{a}^\star, \boldsymbol{w}^{(1)\star}, \boldsymbol{w}^{(2)\star})$ such that $y_i = \boldsymbol{w}^{(2)\star T}\phi(\langle \boldsymbol{w}^{(1)\star}, vec(\boldsymbol{w}^V \boldsymbol{x}_i \boldsymbol{a}_i^{self(z)}))\rangle) + \epsilon_i$, where $\boldsymbol{a}^{self}$ is calculated by (1); $\epsilon_i \sim subG(0, C_4^2)$ for $i = 1, 2, \ldots n$, with $\boldsymbol{x}_i \perp\!\!\!\perp \epsilon_i$. And the output $y$ is centered as (A6).

These assumptions are all parallel to (A3) to (A6).

## A.4 Details about Theorem 3

We consider a $D$-layer network with self-attention structure. We denote the $k^{th}$ self-attention layer follows $g_k(\boldsymbol{x}_g^{k-1}) = \boldsymbol{w}^{k_2}\phi(\langle \boldsymbol{w}^{k_1}, \boldsymbol{w}^V \boldsymbol{x}_g^{k-1}\boldsymbol{a}^{self}\rangle))$, where $\boldsymbol{x}_g^{k-1}$ is the output of $(k-1)^{th}$ self-attention layer, with $\boldsymbol{w}^V \in \mathbb{R}^{d_v \times t}$, $\boldsymbol{x}_g^{k-1} \in \mathbb{R}^{t \times d_{k-1}}$, $\boldsymbol{a}^{self} \in \mathbb{R}^{d_{k-1} \times d_{k-1}}$, $\boldsymbol{w}^{k_1} \in \mathbb{R}^{d_v \times q_k}$ and $\boldsymbol{w}^{k_2} \in \mathbb{R}^{d_k \times q_k}$. $\boldsymbol{a}^{self}$ is calculated in the same way with two-layer self-attention networks. Then we have the final output $h(\boldsymbol{x}) = \boldsymbol{w}^{D_2}\phi(\langle \boldsymbol{w}^{D_1}, vec(\boldsymbol{w}^V \boldsymbol{x}_g^{D-1}\boldsymbol{a}^{self})\rangle)$, where $\boldsymbol{x}_g^{D-1} = (g_{D-1}(\cdots g_1(\boldsymbol{x})))$, and $\boldsymbol{w}^{D_1} \in \mathbb{R}^{1 \times d_v}$, $\boldsymbol{w}^{D_2} \in \mathbb{R}^{d_k}$. In this way, the network calculates self-attention $D$ times and finally produce the final prediction. It is worth mentioning that, to obtain a scalar prediction in regression model, we flatten the value matrix of the last layer as in the two-layer model. We still denote $u = \boldsymbol{w}^{D_2}\phi(\langle \boldsymbol{w}^{D_1}, vec(\boldsymbol{w}^V \boldsymbol{x}_g^{D-1}\boldsymbol{a}^{self})\rangle) - \mathbb{E}(y|\boldsymbol{x})$. Then the necessary assumptions parallel to (A2) are as follows:

(A6) There exists integer $k$ and $r$ such that $k \in \{1, \ldots, D\}$ and $r \in \{1, 2\}$, such that $cov(\nabla h(\boldsymbol{w}^{k_r}), u) \geq c\gamma^2)$ for some constant $c$, and such that $\|\nabla h(\boldsymbol{w}^{k_r})\|_2 \leq c_k$.

This Theorem also requires mild regularity conditions as follows:

- (A3.1) All weights $\boldsymbol{w}^{k_j}$ for $k = 1, \ldots, D$ and $j = 1, 2$ satisfy $\|\boldsymbol{w}^{k_j}\|_2 \leq C_{10}$. And we assume the prediction is centered, i.e. $E(u) = 0$.

- (A3.2) There exists a set of parameters $(\boldsymbol{w}^{(1)\star}, \boldsymbol{w}^{(2)\star})$ such that $y_i = \boldsymbol{w}^{D_2}\phi(\langle \boldsymbol{w}^{D_1}, vec(\boldsymbol{w}^V \boldsymbol{x}_g^{D-1}\boldsymbol{a}^{self})\rangle) + \epsilon_i$ as defined, where $\epsilon_i \sim subG(0, C_4^2)$ for $i = 1, 2, \ldots n$, with $\boldsymbol{x}_i \perp\!\!\!\perp \epsilon_i$. And the output $y$ is centered as (A6).

The following theorem provides sample complexity bound of multi-layer self-attention model:

**Theorem 3** *Under (A6) and regularity assumptions (A3.1) and (A3.2), $d_{self}$ is the total number of parameters in all value, query, key matrices. Then for any $\gamma > 0$, given the sample size: $n \gtrsim \log(\frac{c_k}{\gamma})(d_{self} + D(d_k + d_v)q_k)$, where $c_k$ is the Lipschitz constant of $\nabla h(\boldsymbol{w}^{k_r})$. With probability tending to 1, any stationary point $(\tilde{\boldsymbol{w}}^{(1)}, \tilde{\boldsymbol{w}}^{(2)}, \tilde{\boldsymbol{w}}^Q, \tilde{\boldsymbol{w}}^K)$ satisfies that: $\mathbb{E}(h(\boldsymbol{x}) - \mathbb{E}(y|\boldsymbol{x}))^2 \lesssim \gamma^2$.*

Because multi-layer self-attention models include a large parameter set with complicated gradients, the assumptions are not as intuitive as the two-layer model. But the main assumptions are parallel, such that in an overparameterized network, $\nabla h(\boldsymbol{w})$ spans almost all directions, and some of them are correlated with the bias term $u$. The results show that mainly the size of network parameters determines the sample complexity bound. It implies that an efficient architecture design is critical in reducing the sample complexity. For multi-layer cases, the regular assumptions are stated as following (A3.1) and (A3.2).

## B  RECURRENT ATTENTION MODEL

In this section, we consider analyzing the sample complexity bound for representative recurrent attention framework in Bahdanau et al. (2014). In the recurrent attention network, we follow the setting in self-attention model, such that $\boldsymbol{x}_i = (\boldsymbol{x}_i^1, \ldots, \boldsymbol{x}_i^p) \in \mathbb{R}^{t \times p}$, corresponding to $p$ words with $t$-dimensional embedding. And $\boldsymbol{x}$ is the population version of $\boldsymbol{x}_i$. Then the generative model can be represented as:

$$y_i = \boldsymbol{w}^{(2)T} \langle \boldsymbol{w}^{(1)}, \sum_{j=1}^{p} \boldsymbol{a}(\boldsymbol{x}_i)_j \boldsymbol{x}_i^j \rangle + \epsilon_i$$

Analogous to NLP setting, $\boldsymbol{a}(\boldsymbol{x}_i)$ is a unknown function mapping $\boldsymbol{x}_i$ to a $t$-dimensional vector, where $\boldsymbol{a}(\boldsymbol{x}_i)_j$ represents the effect of the $j^{th}$ word in the sentence for point $i$. Then following the RNN setup in Bahdanau et al. (2014), using data features themselves as their annotations, then for time stamp $k = 1, \ldots, T$, The recurrent attention model estimates for $k^{th}$ time stamp $\boldsymbol{a}^k(\boldsymbol{x}_i)$ as follows:

$$s_k = \boldsymbol{f}(s_{k-1}, c_{k-1}); \ e_{kj} = score(s_{k-1}, \boldsymbol{x}_i^j)$$

$$\alpha_{kj} = \frac{e_{kj}}{\sum_{j=1}^{p} e_{kj}}; \ c_k = \sum_{j=1}^{p} \alpha_{kj} \boldsymbol{x}_i^j$$

$$y_i^{(k)} = \boldsymbol{w}^{(2)T} \phi(\langle \boldsymbol{w}^{(1)}, c_k \rangle)$$

where $score(\cdot)$ is the score function representing how well the inputs around position $j$ and the output at position $i$ match. It can either be a dot product or a MLP. $y_i^{(k)}$ is the prediction in $k^{th}$ time stamp. And we denote $a(\boldsymbol{x}_i)_j = \alpha_{Tj}$, as the attention mask for the final time stamp. And $\boldsymbol{f}(\cdot)$ is the function to update $s_k$. Suppose the parameter set inside these two functions are $\boldsymbol{w}_a$ and $\boldsymbol{w}_f$ with number of parameters as $d_a$ and $d_f$ accordingly. Here we show that when these two functions are expressive enough, recurrent attention networks also have sample complexity bound parallel to self-attention models. Here we introduce necessary assumptions.

(A7) When $T$ is sufficiently large, the output $y$ can be predicted by the two-layer network with an independent sub-Gaussian error with variance $\sigma^2$, i.e, there exists a set of parameters $(\boldsymbol{w}^{(1)\star}, \boldsymbol{w}^{(2)\star})$ such that $y_i = \boldsymbol{w}^{(2)\star T} \phi(\langle \boldsymbol{w}^{(1)\star}, \sum_{j=1}^{p} a(\boldsymbol{x}_i)_j \boldsymbol{x}_i^j \rangle) + \epsilon_i$, where $\epsilon_i \sim subG(0, C_4^2)$ for $i = 1, 2, \ldots n$, with $\boldsymbol{x}_i \perp\!\!\!\perp \epsilon_i$.

(A8) Suppose (A2) holds when we substitute $\boldsymbol{x} \odot \boldsymbol{f}(\boldsymbol{x})$ with $\sum_{j=1}^{p} a(\boldsymbol{x}_i)_j \boldsymbol{x}_i^j$.

(A9) We assume $\|\boldsymbol{w}_a\|_2 \leq C_8$ and $\|\boldsymbol{w}_f\|_2 \leq C_9$.

(A7) to (A9) are parallel to the assumptions in the self-attention case. They can be justified similar to them, which is discussed in Section B of Appendix. When $T$ is large, recurrent attention models can represent a wide class of attentions weights. Thus we assume $y_i$ can be expressed by such recurrent attention models.

Now we can provide the following sample complexity bound.

**Theorem 6.** *Under (A7) to (A9), for any $\gamma > 0$, suppose*

$$n \gtrsim \frac{C_1^2 C_x^2 \eta^2}{\gamma^2} \log(\frac{\eta C_8 C_9}{\gamma})(td + d + d_a + d_f)$$

*where $\eta = C_1 C_2 C_x$, such that if there exist stationary point(s), then with probability tending to 1, any stationary point $(\tilde{\boldsymbol{w}}^{(1)}, \tilde{\boldsymbol{w}}^{(2)}, \tilde{\boldsymbol{w}}_f, \tilde{\boldsymbol{w}}_a)$ satisfies the following prediction error bound:*

$$\mathbb{E}(\tilde{\boldsymbol{w}}^{(2)} \phi(\langle \tilde{\boldsymbol{w}}^{(1)}, \sum_{j=1}^{p} a(\boldsymbol{x}_i)_j \boldsymbol{x}_i^j \rangle) - \mathbb{E}(y|\boldsymbol{x}))^2 \lesssim \gamma^2$$

Theorem 6 provides a sample complexity bound for recurrent attention networks. The bound holds under expressiveness assumption (A7). It also shows a trade-off of expressiveness and sample complexity. When the number of parameters is too large, the sample complexity will be too large; When the number of the parameters is too small, we don't have enough expressiveness to achieve consistency. If $\boldsymbol{f}(\cdot)$ and $\boldsymbol{a}(\cdot)$ are properly selected, they will be sufficiently expressive to obtain good stationary points, and also the number of parameters $d_w$ and $d_f$ will not be too large. In this way, an ideal sample complexity bound to these good stationary points can be achieved as Theorem 7 says. However, with an over-complicated design in these functions, the sample complexity bound will be large; With an over-simple design, such good stationary points don't exist. It is parallel to a trade-off between approximation error and estimation error in learning theory. The theory implies a good design of the recurrent structure will help achieve an optimal sample complexity in recurrent attention model.

## C   APPENDIX FOR SECTION 4

### C.1   DISCUSSION OF THEOREM 4

For illustration, the linear region bounds under different sparsity levels are plotted in Figure 1. The top red line is for baseline model with $p = 100$, and other lines are bounds for attention model with different sparsity level $s_0$.

In a sufficiently wide network, $\lfloor \frac{n_1}{s_0} \rfloor^{s_0}$ is much smaller than $\lfloor \frac{n_1}{p} \rfloor^p$. Then, $(\frac{n_1}{s_0})^{s_0} \leq (\frac{n_1}{p})^p$ holds as long as $n_1 \geq \exp(\frac{p \log p - s_0 \log s_0}{p - s_0})$. Given $\frac{p \log p - s_0 \log s_0}{p - s_0} \leq \frac{p}{p - s_0} \log p$, since $\frac{p}{p - s_0}$ is close to $p$ when $s_0$ is relatively small, the result still holds when $n_1$ is larger than the order of $p$. For illustration, the linear region bounds under different sparsity levels are plotted in Figure 1. The top red line is for baseline model with $p = 100$, and other lines are bounds for attention model with different sparsity level $s_0$. In general, we see the bound of attention model is smaller than the one of baseline model. The result implies that, when appropriate attention mechanism is used, the reduction of number of linear regions leads to a simpler landscape, yet the approximation error remains small. We interpret that attention mechanisms help us reduce unnecessary non-linearity inside the landscape.

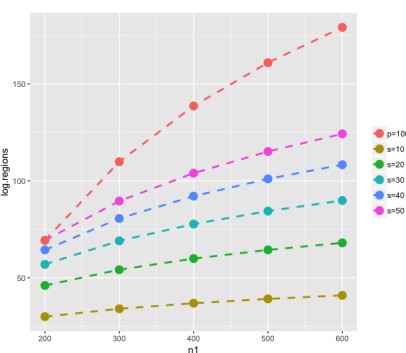

Figure 3: Number of linear regions in log scale v.s. sparsity

### C.2   DISCUSSION OF THEOREM 5

Theorem 5 indicates that property on flatness of minima is maintained when attention mechanism is applied, and there exist good sharp minima, coinciding with the observation in Dinh et al. (2017). However, there is no guarantee that all sharp minima are good in generalization. Revisiting our analysis in Section 3, the restriction on the parameter space helps remove these sharp minima. Specifically, we provide upper bounds on the $\ell_2$ norm of $(\boldsymbol{w}^{(1)}, \boldsymbol{w}^{(2)})$. These constraints restrict the parameter space and remove all sharp minima which we construct in the proof of Theorem 5. In these constructed sharp minima, $\alpha_1$ or $\alpha_2$ goes to infinity, and $\ell_2$ norm bound guarantee that

it cannot happen. In these constructed sharp minima, $\ell_2$ norm bound guarantees that it cannot happen. Practically, $\ell_2$ norm bounds can be achieved through a proper $\ell_2$ regularization, which will be discussed in Section 5. It also coincides with our theoretical and empirical result that a flat minimum are expected to generalize better in general (Keskar et al., 2016).

## C.3 SMALL SAMPLE SIZE

In this section, we study the local minimum of wide neural networks in small sample regime. (Nguyen & Hein, 2017b) proves that a two-layer neural network model can always achieve perfect empirical estimation error when the sample size is small. Here, we extend this result for self-attention model.

**Theorem 7.** *For self-attention model in Sec 3.2, if $rank(\phi(\langle \tilde{\boldsymbol{w}}^{(1)}, vec(\boldsymbol{w}^V \boldsymbol{x}_i \boldsymbol{a}_i^{self}) \rangle)_{i=1,2,...n}) = n$. Then every stationary point $(\tilde{\boldsymbol{w}}^{(1)}, \tilde{\boldsymbol{w}}^{(2)}, \tilde{\boldsymbol{w}}^V, \tilde{\boldsymbol{w}}^Q, \tilde{\boldsymbol{w}}^K)$ of object function (4) is a global minima.*

$rank(\phi(\langle \tilde{\boldsymbol{w}}^{(1)}, \boldsymbol{x}_i \odot \boldsymbol{a} \rangle)_{i=1,2,...n}) = n$ is a mild assumption in a wide network with over-parameterization. We can see that as long as we choose the number of units $d$ to be larger than $n$, the linear dependence of $\langle \boldsymbol{w}^{(1)}, \boldsymbol{x}_i \odot \boldsymbol{a} \rangle_{i=1,2,...n}$ holds with measure zero. In other words, almost surely this matrix has full column rank $n$. Thus after the nonlinear activation, the full column rank still holds almost surely. This assumption is similar to the condition in Theorem 3.8 of Nguyen & Hein (2017b), where they assume the number of units in some layer is larger than the sample size. When the sample size is smaller than the number of units in the network, this theorem holds for the network without attention. It has been proved by Nguyen & Hein (2017b) and Soudry & Carmon (2016) under different conditions.

## D  LEMMA: MODELS WITH A FIXED ATTENTION MASK

In this section, we introduced a attention model with fixed attention mask, as a fundamental building block of attention mechanisms. In the fixed-mask attention model, we consider a dataset $D = \{\boldsymbol{x}_i, y_i\}_{i=1}^N$, $\boldsymbol{x}_i \in \mathbb{R}^p$, $y_i \in \mathbb{R}$, where the output $y_i$ depends only on certain regions of input $\boldsymbol{x}_i$, i.e., $y_i = f^\star(\boldsymbol{a}^\star \odot \boldsymbol{x}_i)$, where $\boldsymbol{a}^\star$ is an unknown fixed attention mask, and $f^\star(.)$ is the ground-truth function that is used to generate the dataset and the vector $\boldsymbol{a}^\star \in [0,1]^p$. The set of entries $\{a_i^\star | a_i^\star \neq 0\}$ corresponds to the relevant region of the input, while the complementary set $\{a_i^\star | a_i^\star = 0\}$ corresponds to the irrelevant region.

The following Lemma 1 on the sample complexity of attention model with fixed attention mask, is not only an important building block of the proof of Theorem 1 but also provides helpful insights for understanding attention mechanisms and revealing the main idea of our proof. With the sparsity structure of attention mask $\boldsymbol{a}$, attention mechanisms constrain parameters in a smaller space, thus they reduce the variability of the empirical landscape, and also reduce the covering number of parameter space. These results lead to a lower sample complexity compared with the baseline model not employing attention. Similar to Corollary 1 result, it is straightforward to calculate the sample complexity bound for the baseline model(not employing attention). To achieve the same error bound, we substitute $s_0$ with $p$ in the bound, and this results in a much larger sample complexity bound.

The attention model we use can be written as:

$$f(\boldsymbol{x}) = \boldsymbol{w}^{(2)T} \phi(\langle \boldsymbol{w}^{(1)}, \boldsymbol{x} \odot \boldsymbol{a} \rangle) \tag{6}$$

The assumptions for analyzing this model is the same as the assumptions of Theorem 1(A1 to A3 and A1.1 to A1.3), where we substitute $\boldsymbol{f}(\boldsymbol{x})$ by $\boldsymbol{a}$ in all assumptions with $\|\boldsymbol{a}\|_0 \leq s_0$. For clarity, we state them here explicitly:

(A.L.1) There exists a set of parameters $(\boldsymbol{w}^{(1)\star}, \boldsymbol{w}^{(2)\star})$ such that $y_i = \boldsymbol{w}^{(2)\star T} \phi(\langle \boldsymbol{w}^{(1)\star}, \boldsymbol{x}_i \odot \boldsymbol{a} \rangle) + \epsilon_i$ for $i = 1, 2, ...n$, where $\epsilon_i$'s follow the sub-Gaussian distribution $subG(0, C_3^2)$ and $\boldsymbol{x}_i \perp\!\!\!\perp \epsilon_i$.

(A.L.2) $\|\boldsymbol{a}\|_0 \leq s_0$ with at most $s_0$ non-zero weights.

(A.L.3) For any $\gamma > 0$, $\lambda_{\max}(Var(\phi(\langle \boldsymbol{w}^{(1)\star}, \boldsymbol{x}_i \odot \boldsymbol{a}^\star \rangle)$
$|\phi(\langle \boldsymbol{w}^{(1)}, \boldsymbol{x} \odot \boldsymbol{a} \rangle)) = o(\gamma^2)$ as $n \gtrsim \frac{k^2}{\gamma^2} \log(\frac{k}{\gamma})(pd + d)$ and $n \to \infty$.

(A.L.4) $\boldsymbol{x}_i$ are i.i.d and $\|\boldsymbol{x}_i\|_\infty < C_x$ for any $i = 1, \ldots, n$.

(A.L.5) There exist $C_1$, $C_2$, $C_\lambda$ such that $\lambda_{\max}(\mathbf{\Sigma}_\phi) \leq C_\lambda$, $\|\boldsymbol{w}^{(1)}\|_F < C_1$ and $\|\boldsymbol{w}^{(2)}\|_2 < C_2$ for any $\boldsymbol{w}^{(1)}, \boldsymbol{w}^{(2)} \in \mathcal{S}$.

(A.L.6) $\mathbb{E}(\boldsymbol{w}^{(2)}\phi(\langle \boldsymbol{w}^{(1)}, \boldsymbol{x} \odot \boldsymbol{a}\rangle)) = \mathbb{E}(y) = 0$.

**Lemma 1.** *Under (A.L.1) to (A.L.6), then for any $\gamma > 0$ and $s_0$ such that $k \lesssim O(s_0)$, suppose*

$$n \gtrsim \frac{s_0^2 C_1^2 C_x^2 \eta^2}{\gamma^2} \log(\frac{s_0 \eta}{\gamma})(pd + p + d)$$

*where $\eta = C_1 C_2 C_x$. Then with probability tending to 1, any stationary point $(\tilde{\boldsymbol{a}}, \tilde{\boldsymbol{w}}^{(1)}, \tilde{\boldsymbol{w}}^{(2)})$ of the objective function (4) satisfies the following prediction error bound:*

$$\mathbb{E}(\tilde{\boldsymbol{w}}^{(2)T}\phi(\langle \tilde{\boldsymbol{w}}^{(1)}, \boldsymbol{x} \odot \tilde{\boldsymbol{a}}\rangle) - \mathbb{E}(y|\boldsymbol{x}))^2 \lesssim \gamma^2 \tag{7}$$

# E    PROOFS

In this section, we first provide a proof sketch for Theorem 1 for readers to understand the high-lelvel idea. Then we provide proofs of all our theorems. Lemma 1, Theorem 1, Theorem 2, Theorem 3 and Theorem 7 are proved in similar manners. For Theorem 1, Theorem 2 and Theorem 3, we omit the exact same part of the proof as Lemma 1.

## E.1    PROOF SKETCH OF THEOREM 1

**Proof Sketch:** We target at proving that under our assumptions and required sample size, all local minimum must have prediction ability as good as global minimum up to $\gamma^2$. The proof can be divided into two major steps. First, we show that for all parameter sets with bad prediction, their $\phi(\boldsymbol{w}^{(1)}, \boldsymbol{x}_i \odot \boldsymbol{a})$ term must be correlated with the bias $E(y|\boldsymbol{x}_i) - E(\boldsymbol{w}^{(2)})\phi(\boldsymbol{w}^{(1)}, \boldsymbol{x}_i \odot \boldsymbol{a})$. It leads to a large magnitude of population gradient with respect to $\boldsymbol{w}^{(2)}$. Second, we construct an $\epsilon$-cover of parameter sets $(\boldsymbol{w}^{(1)}, \boldsymbol{w}^{(2)})$ to show that sample gradients converge to population gradient, such that sample gradient with respect to $\boldsymbol{w}^{(2)}$ is also away from zero. Thus these parameter sets with bad prediction cannot be local minimum. And we conclude that all local minimum must have prediction as good as global minimum up to $O(\gamma^2)$. The complete proof is provided in the Appendix Section E.

## E.2    PROOF OF LEMMA 1

*Proof.* As described in the proof intuition, the proof is divided into two steps. First we study the lower bound of population risk gradient $\|\mathbb{E}_{x,y}(\nabla R_n(\boldsymbol{w}^{(2)}))\|_2$; In step 2 we study the convergence of $\|\nabla R_n(\boldsymbol{w}^{(2)})\|_2$ to the population gradient $\|\mathbb{E}_{x,y}(\nabla R_n(\boldsymbol{w}^{(2)}))\|_2$. We further separate these two steps into three lemmas.

Lemma 1.1 proves the first step, we study the landscape of population risk, showing that with high probability, we know the population risk with respect to $\boldsymbol{x}$ and $\boldsymbol{y}$ $\|\mathbb{E}_{x,y}(\nabla R_n(\boldsymbol{w}^{(2)}))\|_2$ is large; Lemma 1.2 and Lemma 1.3 prove the second step. Specifically, in part (b), we consider the convergence of population risk only taking expectation on $\boldsymbol{y}$, $\mathbb{E}_y(\nabla R_n(\boldsymbol{w}^{(2)}))$. In part (c), finally we consider the convergence of empirical risk gradient $\nabla R_n(\boldsymbol{w}^{(2)})$ to $\mathbb{E}_y(\nabla R_n(\boldsymbol{w}^{(2)}))$.

We introduce necessary notations beforehand. To emphasize the role of $\boldsymbol{x}$ and $\boldsymbol{y}$ separately, here we denote $R(\boldsymbol{w}^{(1)}, \boldsymbol{w}^{(2)}, \boldsymbol{a}) = \mathbb{E}_y(R_n(\boldsymbol{w}^{(1)}, \boldsymbol{w}^{(2)}, \boldsymbol{a}))$, which is the expectation of the empirical loss gradient with respect to $\boldsymbol{y}$, treating $\boldsymbol{x}$ as random, and $\nabla R(\boldsymbol{w})$ as corresponding derivatives. And we denote $\mathbb{E}_x(\nabla R(\boldsymbol{w}^{(1)}, \boldsymbol{w}^{(2)}, \boldsymbol{a})) = \mathbb{E}_{x,y}(\nabla R_n(\boldsymbol{w}^{(1)}, \boldsymbol{w}^{(2)}, \boldsymbol{a}))$, which is the expectation of the empirical loss function with expectation to both $\boldsymbol{x}$ and $\boldsymbol{y}$. In the proof, we may use $o(\gamma)$ for vector/matrix case. In these cases, it means that every element in vector/matrix is $o(\gamma)$.

**Lemma 1.1** Under the assumption of Lemma 1, when equation 7 is violated, with probability going to 1 that, the population risk gradient with respect to $\boldsymbol{w}^{(2)}$ satisfies that:

$$\|\mathbb{E}_{x,y}(\nabla R_n(\boldsymbol{w}^{(2)}))\|_2 \geq O(\gamma)$$

*Proof:* We denote:

$$u = \boldsymbol{w}^{(2)}\phi(\langle \boldsymbol{w}^{(1)}, \boldsymbol{x} \odot \boldsymbol{a}\rangle) - \mathbb{E}(y|\boldsymbol{x}) \tag{8}$$

and $u_i$ as the version with the specified sample index. Then the derivatives of population risk with the expectation to $\boldsymbol{y}$ can be presented as follows:

$$\nabla R(\boldsymbol{w}^{(2)}) = \frac{1}{n}\sum_{i=1}^{n} u_i \phi(\langle \boldsymbol{w}^{(1)}, \boldsymbol{x}_i \odot \boldsymbol{a}\rangle)$$

$$\nabla R(\boldsymbol{w}^{(1)}) = \frac{1}{n}\sum_{i=1}^{n} u_i(\boldsymbol{x}_i \odot \boldsymbol{a})(\boldsymbol{w}^{(2)} \odot \phi^{'}(\langle \boldsymbol{w}^{(1)}, \boldsymbol{x}_i \odot \boldsymbol{a}\rangle))^T$$

$$\nabla R(\boldsymbol{a}) = \frac{1}{n}\sum_{i=1}^{n} u_i(\boldsymbol{x}_i \odot (\boldsymbol{w}^{(1)T}(\boldsymbol{w}^{(2)} \odot \phi^{'}(\langle \boldsymbol{w}^{(1)}, \boldsymbol{x}_i \odot \boldsymbol{a}\rangle))))$$

By (A.L.1), we know that $\mathbb{E}(y_i|\boldsymbol{x}_i) = \boldsymbol{w}^{(2)\star T}\phi(\langle \boldsymbol{w}^{(1)\star}, \boldsymbol{x}_i \odot \boldsymbol{a}^\star\rangle)$. Therefore when $(\boldsymbol{a}, \boldsymbol{w}^{(1)}, \boldsymbol{w}^{(2)}) = (\boldsymbol{a}^\star, \boldsymbol{w}^{(1)\star}, \boldsymbol{w}^{(2)\star})$, all the $u_i$'s are zero, and all the gradients with expectations to $\boldsymbol{x}$ and $y$ are zero. Thus for any true set of parameter $(\boldsymbol{a}^\star, \boldsymbol{w}^{(1)\star}, \boldsymbol{w}^{(2)\star})$, they have zero gradient expectations automatically. And the key of our proof is showing that with high probability, any parameter $(\boldsymbol{a}, \boldsymbol{w}^{(1)}, \boldsymbol{w}^{(2)})$ cannot be stationary point if $\mathbb{E}(|\tilde{\boldsymbol{w}}^{(2)}\phi(\langle \tilde{\boldsymbol{w}}^{(1)}, \boldsymbol{x} \odot \tilde{\boldsymbol{a}}\rangle) - \mathbb{E}(y|\boldsymbol{x})|^2) \geq \gamma^2$, because their gradients w.r.t to $\boldsymbol{w}^{(2)}$ must be bounded away from zero.

In the following section, we prove Lemma 1.1 by showing that if $E(u^2) \geq \gamma^2$, we must have $\|E(\nabla R(\boldsymbol{w}^{(1)}))\|_2 \geq O(\gamma)$. We prove it by contradiction.

In this proof, we denote $\boldsymbol{r} = cov(\mathbb{E}(y|\boldsymbol{x}), \phi(\langle \boldsymbol{w}^{(1)}, \boldsymbol{x} \odot \boldsymbol{a}\rangle)) \in \mathbb{R}^d$, and the covariance matrix for $\phi(\langle \boldsymbol{w}^{(1)}, \boldsymbol{x} \odot \boldsymbol{a}\rangle)$ as:

$$(\boldsymbol{\Sigma}_\phi)_{ij} = cov(\phi_i(\langle \boldsymbol{w}^{(1)}, \boldsymbol{x} \odot \boldsymbol{a}\rangle), \phi_j(\langle \boldsymbol{w}^{(1)}, \boldsymbol{x} \odot \boldsymbol{a}\rangle))$$

If we have $\|E(\nabla R(\boldsymbol{w}^{(2)}))\|_2 = o(\gamma)$, i.e.

$$\|cov(u, \phi(\boldsymbol{w}^{(1)}, \boldsymbol{x}_i \odot \boldsymbol{a}))\|_2 = o(\gamma),$$

and plug it into $cov((\boldsymbol{w}^{(2)})^T\phi(\langle \boldsymbol{w}^{(1)}, \boldsymbol{x} \odot \boldsymbol{a}\rangle), \phi(\langle \boldsymbol{w}^{(1)}, \boldsymbol{x} \odot \boldsymbol{a}\rangle)) = \boldsymbol{r} + o(\gamma)$, we have:

$$\|\boldsymbol{\Sigma}_\phi \boldsymbol{w}^{(2)} - \boldsymbol{r}\|_2 = \|cov(\boldsymbol{w}^{(2)})^T\phi(\langle \boldsymbol{w}^{(1)}, \boldsymbol{x} \odot \boldsymbol{a}\rangle) - E(y|\boldsymbol{x}), \phi(\langle \boldsymbol{w}^{(1)}, \boldsymbol{x} \odot \boldsymbol{a}\rangle))\|_2$$
$$= \|cov(u, \phi(\boldsymbol{w}^{(1)}, \boldsymbol{x}_i \odot \boldsymbol{a}))\|_2 = o(\gamma)$$

Since we know the true mean $y$ can be specified by a true set of parameter $(\boldsymbol{w}^{(1)\star}), \boldsymbol{w}^{(2)\star}, \boldsymbol{a}^\star)$, and we denote that the covariance matrix for $\phi(\langle \boldsymbol{w}^{(1)\star}, \boldsymbol{x} \odot \boldsymbol{a}^\star\rangle)$ as:

$$(\boldsymbol{\Sigma}_\phi)_{ij}^\star = cov(\phi_i(\langle \boldsymbol{w}^{(1)\star}, \boldsymbol{x} \odot \boldsymbol{a}^\star\rangle), \phi_j(\langle \boldsymbol{w}^{(1)\star}, \boldsymbol{x} \odot \boldsymbol{a}^\star\rangle))$$

And we denote the cross-covariance matrix of vector $\phi_i(\langle \boldsymbol{w}^{(1)\star}, \boldsymbol{x} \odot \boldsymbol{a}^\star\rangle)$ and vector $\phi_j(\langle \boldsymbol{w}^{(1)}, \boldsymbol{x} \odot \boldsymbol{a}\rangle)$ as:

$$(\boldsymbol{\Sigma}_c)_{ij}^\star = cov(\phi_i(\langle \boldsymbol{w}^{(1)\star}, \boldsymbol{x} \odot \boldsymbol{a}^\star\rangle), \phi_j(\langle \boldsymbol{w}^{(1)}, \boldsymbol{x} \odot \boldsymbol{a}\rangle))$$

Then with zero expectation on $E(y|\boldsymbol{x})$ and $E(\boldsymbol{w}^{(2)T}\phi(\boldsymbol{w}^{(1)}, \boldsymbol{x}_i \odot \boldsymbol{a}))$ we have:

$$E(u^2) = var(E(y|\boldsymbol{x}) - \boldsymbol{w}^{(2)T}\phi(\boldsymbol{w}^{(1)}, \boldsymbol{x}_i \odot \boldsymbol{a}))$$
$$= var(E(y|\boldsymbol{x})) + var(\boldsymbol{w}^{(2)T}\phi(\boldsymbol{w}^{(1)}, \boldsymbol{x}_i \odot \boldsymbol{a})) - 2cov(E(y|\boldsymbol{x}), \boldsymbol{w}^{(2)T}\phi(\boldsymbol{w}^{(1)}, \boldsymbol{x}_i \odot \boldsymbol{a}))$$
$$= var(\boldsymbol{w}^{(2)\star T}\phi(\langle \boldsymbol{w}^{(1)\star}, \boldsymbol{x} \odot \boldsymbol{a}^\star\rangle)) + var(\boldsymbol{w}^{(2)T}\phi(\boldsymbol{w}^{(1)}, \boldsymbol{x}_i \odot \boldsymbol{a}))$$
$$- 2cov(\boldsymbol{w}^{(2)\star T}\phi(\langle \boldsymbol{w}^{(1)\star}, \boldsymbol{x} \odot \boldsymbol{a}^\star\rangle), \boldsymbol{w}^{(2)T}\phi(\boldsymbol{w}^{(1)}, \boldsymbol{x}_i \odot \boldsymbol{a}))$$
$$= \boldsymbol{w}^{(2)\star T}\boldsymbol{\Sigma}_\phi^\star\boldsymbol{w}^{(2)\star} - 2\boldsymbol{w}^{(2)\star T}\boldsymbol{\Sigma}_c^\star\boldsymbol{w}^{(2)} + \boldsymbol{r}^T\boldsymbol{\Sigma}_\phi\boldsymbol{r} + c_\lambda o(\gamma^2)$$
$$= \boldsymbol{w}^{(2)\star T}\boldsymbol{\Sigma}_\phi^\star\boldsymbol{w}^{(2)\star} - 2\boldsymbol{w}^{(2)\star T}\boldsymbol{\Sigma}_c\boldsymbol{\Sigma}_\phi^{-1}\boldsymbol{\Sigma}_c\boldsymbol{w}^{(2)\star} + \boldsymbol{w}^{(2)\star T}\boldsymbol{\Sigma}_c\boldsymbol{\Sigma}_\phi^{-1}\boldsymbol{\Sigma}_c\boldsymbol{w}^{(2)\star} + c_\lambda o(\gamma)$$
$$= \boldsymbol{w}^{(2)\star T}(\boldsymbol{\Sigma}_\phi^\star - \boldsymbol{\Sigma}_c\boldsymbol{\Sigma}_\phi^{-1}\boldsymbol{\Sigma}_c)\boldsymbol{w}^{(2)\star} + c_\lambda o(\gamma^2)$$

where $\boldsymbol{\Sigma}_\phi^\star - \boldsymbol{\Sigma}_c \boldsymbol{\Sigma}_\phi^{-1} \boldsymbol{\Sigma}_c$ is exactly the residual of covariance matrix after taking regression on $\{\phi_{1...d}(\langle \boldsymbol{w}^{(1)}, \boldsymbol{x} \odot \boldsymbol{f}(\boldsymbol{x})\rangle)\}$, as we defined in (A.L.3).(Page 176 of Izenman (2013))

$\boldsymbol{w}^{(2)\star T}(\boldsymbol{\Sigma}_\phi^\star - \boldsymbol{\Sigma}_c \boldsymbol{\Sigma}_\phi^{-1} \boldsymbol{\Sigma}_c)\boldsymbol{w}^{(2)\star}$ is a quadratic form of that conditional variance matrix. By (A.L.5), we know its largest eigenvalue $\lambda_{\max}(\boldsymbol{\Sigma}_\phi^\star - \boldsymbol{\Sigma}_c \boldsymbol{\Sigma}_\phi^{-1} \boldsymbol{\Sigma}_c) = o(\gamma)$. Finally we obtain that:

$$\begin{aligned} E(u^2) &= \|E(\nabla R(\boldsymbol{w}^{(2)}))\|_2 + c_\lambda o(\gamma^2) \\ &\leq \|\boldsymbol{w}^{(2)\star}\|_2^2 * \lambda_{\max}(\boldsymbol{\Sigma}_\phi^\star - \boldsymbol{\Sigma}_c \boldsymbol{\Sigma}_\phi^{-1} \boldsymbol{\Sigma}_c) + c_\lambda o(\gamma^2) \\ &= o(\gamma^2) \end{aligned}$$

By contradiction, we conclude that if $E(u^2) \geq \gamma^2$, we must have $\|E(\nabla R(\boldsymbol{w}^{(2)})\|_2 \geq O(\gamma)$. Here we finish the proof of Lemma 1.1.

**Lemma 1.2** Under the assumption of Lemma 1, when equation 7 is violated, the risk gradient of $\boldsymbol{w}^{(2)}$ with expectation to $y$ satisfies:

$$\|\mathbb{E}_y(\nabla R_n(\boldsymbol{w}^{(2)}))\|_2 \geq O(\gamma)$$

*proof:* In Lemma 1.1 we have shown that in population level, all parameter sets with bad prediction has a large magnitude of $E(\nabla R(\boldsymbol{w}^{(2)}))$. In Lemma 1.2, we use $\epsilon$-cover technique to bound the gap between $\nabla R(\boldsymbol{w}^{(2)})$(recall that $\nabla R(\boldsymbol{w}^{(2)})$ is short notation of $\mathbb{E}_y(\nabla R_n(\boldsymbol{w}^{(2)}))$) and $E(\nabla R(\boldsymbol{w}^{(2)}))$.

Our parameters $\boldsymbol{w}^{(1)}, \boldsymbol{w}^{(2)}$ and $\boldsymbol{a}$ are inside the $\ell_2$ balls $B^d(0, C_1), B^{p \times d}(0, C_2)$ and $B^p(0, s_0^2)$, where $B^p(c, r^2)$ represents a p-dimensional $\ell_2$ with center $c$ and radius $r$. By Lemma 5.2 in Vershynin (2010), the $\epsilon$-covering number $N_{\epsilon_1}, N_{\epsilon_2}, N_{\epsilon_3}$ for these three balls are upper bounded by:

$$N_{\epsilon_1} \leq (3C_1/\epsilon)^d, N_{\epsilon_2} \leq (3C_2/\epsilon)^{pd}, N_{\epsilon_3} \leq (3s_0^2/\epsilon)^p$$

Then we know the joint $3\epsilon$-covering number for the union of all three parameters $N_{3\epsilon}$ satisfies that $N_{3\epsilon} \leq N_{\epsilon_1} N_{\epsilon_2} N_{\epsilon_3}$. For the ease of notation, we denote $\boldsymbol{\theta} = (\boldsymbol{a}, \boldsymbol{w}^{(1)}, \boldsymbol{w}^{(2)})$. Let $\boldsymbol{\Theta}_\epsilon = \{\boldsymbol{\theta}_1, \cdots, \boldsymbol{\theta}_{N_\epsilon}\}$ be a corresponding cover with $N_{3\epsilon}$ elements. Following the $\epsilon$-covering we construct for $(\boldsymbol{w}^{(1)}, \boldsymbol{w}^{(2)}, \boldsymbol{a})$ separately, we can always find $\boldsymbol{\Theta}_\epsilon$ such that for any feasible $\boldsymbol{\theta}$, there exists $j \in [N]$ such that $\max(\|\boldsymbol{w}_{(j)}^{(1)} - \boldsymbol{w}^{(1)}\|_2, \|\boldsymbol{w}_{(j)}^{(2)} - \boldsymbol{w}^{(2)}\|_2, \|\boldsymbol{a}_{(j)} - \boldsymbol{a}\|_2) \leq \epsilon$. In this proof, we use parenthesis subscription $(j)$ to represent the $j^{th}$ element in the cover, to distinguish it from other subscriptions.

By triangle inequality, we have:

$$\|\nabla R(\boldsymbol{w}^{(2)})\|_2 \geq \|\nabla R(\boldsymbol{w}_{(j)}^{(2)})\|_2 - \|\nabla R(\boldsymbol{w}^{(2)}) - \nabla R(\boldsymbol{w}_{(j)}^{(2)})\|_2 \tag{9}$$

Therefore, we only need to bound both parts on the r.h.s of equation 9. We start with the first term $\|\nabla R(\boldsymbol{w}_{(j)}^{(2)})\|_2$. To achieve this, we first bound the gradient term $\boldsymbol{v}_i = u_i \phi(\boldsymbol{w}^{(1)}, \boldsymbol{x}_i \odot \boldsymbol{a})$. For any fixed parameter set, we calculate:

$$\|\boldsymbol{v}_i\|_2^2 \lesssim (C_2 \|\phi(\boldsymbol{w}^{(1)}, \boldsymbol{x}_i \odot \boldsymbol{a})\|_2)^2 \|\phi(\boldsymbol{w}^{(1)}, \boldsymbol{x}_i \odot \boldsymbol{a})\|_2^2$$

Here we denote $\|\boldsymbol{w}^{(1)}\|_{1,active}$ as the $\ell_1$ norm of $\boldsymbol{w}^{(1)}$ on active features, i.e. on the feature such that its attention weight $a$ is not zero, and from the sparsity condition we know that there are at most $s_0$ such nonzero elements in $\boldsymbol{x} \odot \boldsymbol{a}$. Combining with the $\ell_2$ bound of $\boldsymbol{w}^{(2)}$, we apply Cauchy-Schwarz inequality:

$$\|\phi(\boldsymbol{w}^{(1)}, \boldsymbol{x}_i \odot \boldsymbol{a})\|_2 \leq \max\{|\boldsymbol{x} \odot \boldsymbol{a}|\}\|\boldsymbol{w}^{(1)}\|_{1,active} = \sqrt{s_0} C_x C_1$$

It implies:

$$\|\boldsymbol{v}_i\|_2^2 \lesssim (C_2 \|\phi(\boldsymbol{w}^{(1)}, \boldsymbol{x}_i \odot \boldsymbol{a})\|_2)^2 \|\phi(\boldsymbol{w}^{(1)}, \boldsymbol{x}_i \odot \boldsymbol{a})\|_2^2 = O(s_0^2 C_1^4 C_2^2 C_x^4) \tag{10}$$

From Lemma 1.1, we know there exists a constant $c$ such that $\|\mathbb{E}_x(\nabla R(\boldsymbol{w}^{(2)}))\|_2 \geq c\gamma$ for some constant c. We denote $\xi^2 = s_0^2 C_1^4 C_2^2 C_x^4$. Then we apply Hoeffding bound on the $\ell_2$ norm of

$\mathbb{E}_x(\nabla R(\boldsymbol{w}_{(j)}^{(2)}))$, with the upper bound of it as $O(\sigma^2)$. Denoting $\nabla R(\boldsymbol{w}_{(j)}^{(2)})$ as the gradient with respect to $j^{th}$ parameter set in $\epsilon$-cover for $j \in \{1, \dots, N_\epsilon\}$:

$$P(\|\nabla R(\boldsymbol{w}_{(j)}^{(2)}) - \mathbb{E}_x(\nabla R(\boldsymbol{w}_{(j)}^{(2)}))\|_2 \geq \frac{c\gamma}{3}) \lesssim \exp(-n\frac{c^2\gamma^2}{\xi^2})$$

Then by union bound, over all $N_\epsilon$ elements in $\boldsymbol{\Theta}_\epsilon$:

$$P(\exists j \in [N_\epsilon], \|\nabla R(\boldsymbol{w}_{(j)}^{(2)})\|_2 \geq \frac{2c\gamma}{3}) \gtrsim 1 - N_\epsilon \exp(-n\frac{c^2\gamma^2}{\xi^2})$$

Secondly we analyze $\|\nabla R(\boldsymbol{w}^{(2)}) - \nabla R(\boldsymbol{w}_{(j)}^{(2)})\|_2$ term. Here we use $u_i$ to represent the prediction error for $i^{th}$ instant with respect to parameter $(\boldsymbol{a}, \boldsymbol{w}^{(1)}, \boldsymbol{w}^{(2)})$, and use $u_{i(j)}$ to represent the term with respect to the parameter from $j^{th}$ element in $\epsilon$-cover set. By triangle inequality, we have:

$$\|\nabla R(\boldsymbol{w}^{(2)}) - \nabla R(\boldsymbol{w}_{(j)}^{(2)})\|_2 \leq \frac{2}{n}\|\sum_{i=1}^n (u_i\phi(\langle \boldsymbol{w}^{(1)}, \boldsymbol{x}_i \odot \boldsymbol{a}\rangle) - u_{i(j)}\phi(\langle \boldsymbol{w}_{(j)}^{(1)}, \boldsymbol{x}_i \odot \boldsymbol{a}_{(j)}\rangle))\|_2$$

$$\lesssim \frac{2}{n}(\|\sum_{i=1}^n (u_i - u_{i(j)})\phi(\langle \boldsymbol{w}^{(1)}, \boldsymbol{x}_i \odot \boldsymbol{a}\rangle)\|_2 + \|\sum_{i=1}^n u_{i(j)}(\phi(\langle \boldsymbol{w}^{(1)}, \boldsymbol{x}_i \odot \boldsymbol{a}\rangle) - \phi(\langle \boldsymbol{w}_{(j)}^{(1)}, \boldsymbol{x}_i \odot \boldsymbol{a}_{(j)}\rangle))\|_2)$$

$$\lesssim s_0 C_X^2 C_1^2 C_2 \epsilon$$

We choose $\epsilon = \frac{c\gamma}{3s_0 C_x^2 C_1^2 C_2}$, and plug back above results into equation 9, then at least with probability $1 - \mathcal{O}(N_\epsilon \exp(-n\frac{c^2\gamma^2}{\xi^2}))$, we have $\|\nabla R(\boldsymbol{w}^{(2)})\|_2 > \frac{c\gamma}{3}$. Therefore we can choose:

$$n \gtrsim \frac{\sigma^2}{c^2\gamma^2}\log(\frac{s_0 C_1 C_2 C_x}{c\gamma})(pd + p + d)$$

such that $N_\epsilon \exp(-n\frac{c^2\gamma^2}{\xi^2}) = o(1)$. Finally we can conclude that with probability $1 - o_n(1)$, for any $(\boldsymbol{a}, \boldsymbol{w}^{(1)}, \boldsymbol{w}^{(2)})$ such that $\mathbb{E}(\tilde{\boldsymbol{w}}^{(2)}\phi(\langle \tilde{\boldsymbol{w}}^{(1)}, \boldsymbol{x} \odot \tilde{\boldsymbol{a}}\rangle) - \mathbb{E}(y|\boldsymbol{x}))^2 \geq \gamma$, we have $\|\nabla R(\boldsymbol{w}^{(2)})\|_2 > \frac{c\gamma}{3}$.

**Lemma 1.3** Under the assumption of Lemma 1, when equation 7 is violated, with probability going to 1 that, the empirical risk gradient with respect to $\boldsymbol{w}^{(2)}$ satisfies that:

$$\|\nabla R_n(\boldsymbol{w}^{(2)})\|_2 \geq O(\gamma)$$

*Proof:* So far in Lemma 1.2, we have shown that for population risk with respect to $y$, with high probability, all the parameter sets with poor prediction in expectation, i.e $\mathbb{E}(u^2) \geq O(\gamma^2)$, their population risk gradient with expectation to $\boldsymbol{y}$ must be away from zero. Now we move forward to show that empirical risk $\|\nabla R_n(\boldsymbol{w}^{(2)})\|_2$ converges to $\|\nabla R(\boldsymbol{w}^{(2)})\|_2$. In aspect of $\boldsymbol{w}^{(2)}$, they can be represented as:

$$\nabla R_n(\boldsymbol{w}^{(2)}) - \nabla R(\boldsymbol{w}^{(2)}) = \frac{1}{n}\sum_{i=1}^n \epsilon_i \phi(\langle \boldsymbol{w}^{(1)}, \boldsymbol{x}_i \odot \boldsymbol{a}\rangle)$$

With (A.L.1), we know that $\epsilon_i \sim subG(0, C_3^2)$, thus $\frac{1}{n}\sum_{i=1}^n \epsilon_i = O(\frac{1}{\sqrt{n}})$ by C.L.T, combining the bound for $\phi(\langle \boldsymbol{w}^{(1)}, \boldsymbol{x}_i \odot \boldsymbol{a}\rangle)$ we have derived in the proof of Lemma 1.2, with sample size $n \gtrsim \frac{\xi^2}{c^2\gamma^2}\log(\frac{s_0 C_1 C_2 C_x}{c\gamma})(pd + p + d)$, conclude that with probability $1 - o_p(1)$:

$$\|\nabla R_n(\boldsymbol{w}^{(2)}) - \nabla R(\boldsymbol{w}^{(2)})\|_2 \leq \frac{c\gamma}{6} \tag{11}$$

$$\tag{12}$$

Recalling part (a), under the first case that w.h.p $\|\nabla R(\boldsymbol{w}^{(2)})\|_2 \geq \frac{c\gamma}{3}$ for any parameter $(\boldsymbol{a}, \boldsymbol{w}^{(1)}, \boldsymbol{w}^{(2)})$ with $\|\boldsymbol{w}^{(2)}\phi(\langle \boldsymbol{w}^{(1)}, \boldsymbol{X} \odot \boldsymbol{a}\rangle) - \mathbb{E}(\boldsymbol{y}|\boldsymbol{X})\|_2 \geq \gamma$. Combining this with (11), we conclude that for any positive constant $\gamma > 0$, with required sample size, with high probability that $\|\nabla R_n(\boldsymbol{w}^{(2)})\|_2 > 0$, thus they cannot be stationary solution for our loss function.

In other words, under our assumptions, all the stationary points $(\tilde{\boldsymbol{a}}, \tilde{\boldsymbol{w}}^{(1)}, \tilde{\boldsymbol{w}}^{(2)})$ in our programming satisfy the prediction error upper bound rate $\gamma$ w.h.p, when sample size:

$$n \gtrsim \frac{s_0^2 \eta^2}{c^2 \gamma^2} \log(\frac{s_0 \eta}{c \gamma})(pd + p + d)$$

$\square$

### E.3 Proof of Theorem 1

*Proof.* Theorem 1 can be proved following the same manner as Lemma 1, substituting $\boldsymbol{a}$ by $\boldsymbol{f}(\boldsymbol{x})$, only changing two bounds. Here we specify these two different bounds with Lemma 1. First difference is the bound of $\|\phi(\langle \boldsymbol{w}^{(1)}, \boldsymbol{x}_i \odot \boldsymbol{f}(\boldsymbol{x})\rangle)\|_2$ in Lemma 1.2, and the second is the $\epsilon$-cover number in Lemma 1.2.

By assumption (A2), we denote $\boldsymbol{f}(\boldsymbol{x})_{lead} = \{\boldsymbol{f}(\boldsymbol{x})_{(1)}, \ldots, \boldsymbol{f}(\boldsymbol{x})_{(s_0)}\}$ as the top $s_0$ leading attention weights such that $\|\boldsymbol{f}(\boldsymbol{x})_{lead}\|_2 \geq 1 - \tau$. And $\|\boldsymbol{f}(\boldsymbol{x})_{sub}\|$ as the other $p - s_0$ attention weights and it satisfies $\|\boldsymbol{f}(\boldsymbol{x})_{sub}\|_2 \leq \tau$ as $\|\boldsymbol{f}(\boldsymbol{x})\|_2 = \|\boldsymbol{f}(\boldsymbol{x})_{lead}\|_2 + \|\boldsymbol{f}(\boldsymbol{x})_{sub}\|_2 = 1$. We denote $\boldsymbol{x}_{lead}$ and $\boldsymbol{w}_{lead}^{(1)}$ as the features and network weights corresponding to leading attention weights, and $\boldsymbol{x}_{sub}$, $\boldsymbol{w}_{sub}^{(1)}$ as the features and network weights corresponding to other attention weights.

Parallel to Lemma 1.2, we just need to adjust the bound of $\phi(\langle \boldsymbol{w}^{(1)}, \boldsymbol{x}_i \odot \boldsymbol{f}(\boldsymbol{x})\rangle)$ term as:

$$\|\phi(\langle \boldsymbol{w}^{(1)}, \boldsymbol{x}_i \odot \boldsymbol{f}(\boldsymbol{x})\rangle)\|_2 = \|\phi(\langle \boldsymbol{w}_{lead}^{(1)}, \boldsymbol{x}_{lead} \odot \boldsymbol{f}(\boldsymbol{x})_{lead}\rangle + \langle \boldsymbol{w}_{sub}^{(1)}, \boldsymbol{x}_{sub} \odot \boldsymbol{f}(\boldsymbol{x})_{sub}\rangle)\|_2$$
$$\leq \sqrt{2}(\|\phi(\langle \boldsymbol{w}_{lead}^{(1)}, \boldsymbol{x}_{lead} \odot \boldsymbol{f}(\boldsymbol{x})_{lead}\rangle)\|_2 + \|\phi(\langle \boldsymbol{w}_{sub}^{(1)}, \boldsymbol{x}_{sub} \odot \boldsymbol{f}(\boldsymbol{x})_{sub}\rangle)\|_2$$

The two terms in the last inequality can be bounded by Cauchy-Schwarz inequality separately:

$$\|\phi(\langle \boldsymbol{w}_{lead}^{(1)}, \boldsymbol{x}_{lead} \odot \boldsymbol{f}(\boldsymbol{x})_{lead}\rangle)\|_2 \leq \|\boldsymbol{w}^{(1)}\|_2 \|\boldsymbol{x}_{lead} \odot \boldsymbol{f}(\boldsymbol{x})_{lead}\|_2$$
$$\leq \|\boldsymbol{w}^{(1)}\|_2 \|\boldsymbol{x}_{lead}\|_2 \|\boldsymbol{f}(\boldsymbol{x})_{lead}\|_2 \leq C_1 * \sqrt{s_0} * C_x * (1 - \tau) \leq \sqrt{s_0} C_1 C_x$$

And

$$\|\phi(\langle \boldsymbol{w}_{sub}^{(1)}, \boldsymbol{x}_{sub} \odot \boldsymbol{f}(\boldsymbol{x})_{sub}\rangle)\|_2 \leq \|\boldsymbol{w}^{(1)}\|_2 \|\boldsymbol{x}_{sub} \odot \boldsymbol{f}(\boldsymbol{x})_{sub}\|_2$$
$$\leq \|\boldsymbol{w}^{(1)}\|_2 \|\boldsymbol{x}_{sub}\|_2 \|\boldsymbol{f}(\boldsymbol{x})_{sub}\|_2 \leq C_1 * \sqrt{p} * C_x * \tau = \tau \sqrt{p} C_1 C_x$$

Combining both inequalities, we have:

$$\|\phi(\langle \boldsymbol{w}^{(1)}, \boldsymbol{x}_i \odot \boldsymbol{f}(\boldsymbol{x})\rangle)\|_2 \leq \sqrt{2}(\sqrt{s_0} + \tau\sqrt{p})C_1 C_x \tag{13}$$

Further we have

$$\|\boldsymbol{v}_i\|_2^2 \lesssim (C_2 \|\phi(\langle \boldsymbol{w}^{(1)}, \boldsymbol{x}_i \odot \boldsymbol{f}(\boldsymbol{x})\rangle)\|_2)^2 \|\phi(\langle \boldsymbol{w}^{(1)}, \boldsymbol{x}_i \odot \boldsymbol{f}(\boldsymbol{x})\rangle)\|_2^2 = (\sqrt{s_0} + \tau\sqrt{p})^4 C_1^4 C_2^2 C_x^4 \tag{14}$$

Second, in this case, since $\boldsymbol{f}(\boldsymbol{x})$ is not optimized together, but calculated from $\boldsymbol{x}$ instead. We don't have to consider the $\epsilon$-cover number for $\boldsymbol{a}$ in the maximum operator. Therefore the new $\epsilon$-cover number for $\boldsymbol{w}^{(1)}$ and $\boldsymbol{w}^{(2)}$ are upper bounded as:

$$N_{\epsilon_1} \leq (3C_1/\epsilon)^d, N_{\epsilon_2} \leq (3C_2/\epsilon)^{pd}$$

Substituting the new $\epsilon$-cover bound to the theorem, we obtain the final sample complexity bound:

$$n \gtrsim \frac{(\sqrt{s_0} + \tau\sqrt{p})^4 \eta^2}{\gamma^2} \log(\frac{s_0 \eta}{\gamma})(pd + d)$$

$\square$

### E.4 Proof of Theorem 2

*Proof.* Theorem 2 can be proved following the same manner as Lemma 1, substituting $\boldsymbol{a}$ by $\boldsymbol{a}^{self}$, only changing several bounds. Here we specify the differences with Lemma 1.

Finally, we obtained a new $\epsilon$-covering bound accordingly:

$$N_{\epsilon_1} \leq (3C_1/\epsilon)^d, N_{\epsilon_2} \leq (3C_2/\epsilon)^{pd_v d}, N_{\epsilon_3} \leq (3C_5/\epsilon)^{td_q}, N_{\epsilon_4} \leq (3C_6/\epsilon)^{td_q}, N_{\epsilon_5} \leq (3C_7/\epsilon)^{td_v},$$

where $N = \Pi_{i=1}^5 N_{\epsilon_i}$.

Then, the new $\boldsymbol{v}_i$ terms parallel to Lemma 1.2 is:

$$\boldsymbol{v}_i = u_i \phi(\langle \boldsymbol{w}^{(1)}, vec(\boldsymbol{w}^V \boldsymbol{x}_i \boldsymbol{a}_i^{self})\rangle)$$

where $u_i = (\boldsymbol{w}^{(2)T}\phi(\langle \boldsymbol{w}^{(1)}, vec(\boldsymbol{w}^V \boldsymbol{x}_i \boldsymbol{a}_i^{self})) - E(y_i|x_i)$. Under assumptions, using the same argument as Lemma 1.1 and Theorem 1 under overparameterized network, we have that, there exists a constant $c$ such that either $\|\mathbb{E}(\nabla R(\boldsymbol{w}^{(2)}))\|_2 \geq c\gamma$.

In the case when $\|\mathbb{E}(\nabla R(\boldsymbol{w}^{(2)}))\|_2 \geq c\gamma$. Here we derive the new bound of $\|\boldsymbol{v}\|_2^2$ with respect to $\boldsymbol{x}$. We have:

$$\|\boldsymbol{v}_i\|_2^2 \lesssim (C_2\|\phi(\langle \boldsymbol{w}^{(1)}, vec(\boldsymbol{w}^V \boldsymbol{x}_i \boldsymbol{a}_i^{self})\rangle)\|_2)^2 \|\phi(\langle \boldsymbol{w}^{(1)}, vec(\boldsymbol{w}^V \boldsymbol{x}_i \boldsymbol{a}_i^{self})\rangle)\|_2^2$$

Then we derive the bound of $\|\phi(\langle \boldsymbol{w}^{(1)}, vec(\boldsymbol{w}^V \boldsymbol{x}_i \boldsymbol{a}_i^{self})\rangle)\|_2$. Using our assumption (A4): $\|\boldsymbol{a}_{i(lead)}^{self}\|_2 \geq 1 - \tau$, following the same steps as derived in equation 13, we define $\boldsymbol{a}_{lead}^{self}$ and $\boldsymbol{a}_{sub}^{self}$ as top $s_0$ leading attention weights and other following attention weights. Then we have:

$$
\begin{aligned}
\|\phi(\langle \boldsymbol{w}^{(1)}, vec(\boldsymbol{w}^V \boldsymbol{x}_i \boldsymbol{a}_i^{self})\rangle)\|_2 &\leq \|\boldsymbol{w}^{(1)}\|_2 \|\boldsymbol{w}^V\|_2 \|\boldsymbol{x}_i \odot \boldsymbol{a}_i^{self}\|_2 \\
&\leq \sqrt{2}C_1 C_8(\|\langle \boldsymbol{x}_{lead} \odot \boldsymbol{a}_{lead}^{self}\rangle\|_2 + \|\boldsymbol{x}_{sub} \odot \boldsymbol{a}_{sub}^{self}\rangle\|_2) \\
&\leq \sqrt{2}(\sqrt{s_0} + \tau\sqrt{p})C_1 C_8
\end{aligned}
$$

Finally we obtain:

$$\|\boldsymbol{v}_i\|_2^2 \lesssim (C_2\|\phi(\langle \boldsymbol{w}^{(1)}, vec(\boldsymbol{w}^V \boldsymbol{x}_i \boldsymbol{a}_i^{self})\rangle)\|_2)^2 \|\phi(\langle \boldsymbol{w}^{(1)}, vec(\boldsymbol{w}^V \boldsymbol{x}_i \boldsymbol{a}_i^{self})\rangle)\|_2^2 = 4(\sqrt{s_0} + \tau\sqrt{p})^4 C_1^4 C_2^2 C_8^4 C_x^4$$

Denoting $\xi^2 = (\sqrt{s_0} + \tau\sqrt{p})^4 C_1^2 C_8^2 C_x^2$, parallel to Theorem 3, applying Hoeffding bound and union bound:

$$P(\exists j \in [N_\epsilon], \|\boldsymbol{v}_j\|_2 \leq \frac{2c\gamma}{3}) \lesssim N_\epsilon \exp(-n\frac{c^2\gamma^2}{\xi^2})$$

Second, we bound $\|\nabla R(\boldsymbol{w}^{(2)}) - \nabla R(\boldsymbol{w}^{(2)})_{(j)}\|_2$ term. The attention weight gap is bounded as:

$$
\begin{aligned}
\|\boldsymbol{Q}_i^z \boldsymbol{K}_i^T - \boldsymbol{Q}_{i(j)}^z \boldsymbol{K}_{i(j)}^T\|_{\max} &= \|(\boldsymbol{x}_i^z)^T((\boldsymbol{w}^Q)^T \boldsymbol{w}^k)\boldsymbol{x}_i - (\boldsymbol{x}_i^z)^T(\boldsymbol{w}_{(j)}^Q \boldsymbol{w}_{(j)}^k)\boldsymbol{x}_i\|_{\max} \\
&\lesssim t(C_5 + C_6)C_x^2 \epsilon
\end{aligned}
$$

With this bound, we know $\boldsymbol{a}_i^{self(z)}$ is Lipschitz function under assumption (A9) such that $\|\boldsymbol{Q}_i^z \boldsymbol{K}_i^T\|_2$ has lower bounded. Recall the softmax function for vector $\boldsymbol{\beta}$ is defined as:

$$softmax(\boldsymbol{\beta}) = \frac{exp(\boldsymbol{\beta})}{\sum_{i=1}^p exp(\boldsymbol{\beta}_i)}$$

It is Lipschitz continuous on $\boldsymbol{\beta}$ when $\sum_{i=1}^p exp(\boldsymbol{\beta}_i)$ has a lower bound, which is satisfied by (A2.1). Thus the bound can be derived as:

$$
\begin{aligned}
\|\boldsymbol{a}_i^{self(z)} - \boldsymbol{a}_{i(j)}^{self(z)}\|_2 &\lesssim \|softmax(\frac{\boldsymbol{Q}_i^z \boldsymbol{K}_i^T}{\sqrt{d_k}}) - softmax(\frac{\boldsymbol{Q}_{i(j)}^z \boldsymbol{K}_{i(j)}^T}{\sqrt{d_k}})\|_2 \\
&\lesssim \sqrt{p}t(C_5 + C_6)C_x^2 \epsilon
\end{aligned}
$$

With these bounds, we have:

$$\|\nabla R(\boldsymbol{w}^{(2)}) - \nabla R(\boldsymbol{w}^{(2)})_j\|_2$$

$$\leq \frac{1}{n}\|\sum_{i=1}^{n} u_{i(j)}\phi(\langle \boldsymbol{w}^{(1)}, vec(\boldsymbol{w}^V \boldsymbol{x}_i \boldsymbol{a}_i^{self})\rangle) - u_{i(j)}\phi(\langle \boldsymbol{w}_{(j)}^{(1)}, vec(\boldsymbol{w}^V \boldsymbol{x}_i \boldsymbol{a}_{i(j)}^{self})\rangle)\|_2$$

$$\lesssim \frac{1}{n}(\|\sum_{i=1}^{n} (u_i - u_{i(j)})\phi(\langle \boldsymbol{w}^{(1)}, vec(\boldsymbol{w}^V \boldsymbol{x}_i \boldsymbol{a}_i^{self}))\|_2$$

$$+ \|\sum_{i=1}^{n} u_{i(j)}(\phi(\langle \boldsymbol{w}^{(1)}, \boldsymbol{w}^V \boldsymbol{x}_i \boldsymbol{a}_i^{self}) - u_{i(j)}\phi(\langle \boldsymbol{w}_{(j)}^{(1)}, vec(\boldsymbol{w}_{(j)}^V \boldsymbol{x}_i \boldsymbol{a}_{i(j)}^{self})))\|_2)$$

$$\lesssim ((\sqrt{s_0} + \tau\sqrt{p})^2 t C_1^2 C_2^2 C_8^2 C_x^2 \epsilon + \sqrt{p} t C_1 C_2 C_8 (C_5 + C_6) C_x^2 \epsilon$$

$$\lesssim ((\sqrt{s_0} + \tau\sqrt{p})^2 + \sqrt{p} t)(C_1 C_2 C_8 (C_1 C_2 C_8 + (C_5 + C_6))\epsilon$$

Recall that $u_{i(j)}$ and $a_{i(j)}$ are corresponding to the $j^{th}$ element in the epsilon cover. Denote $\xi = ((\sqrt{s_0} + \tau\sqrt{p})^2 + \sqrt{p} t)(C_1 C_2 C_8 (C_1 C_2 C_8 + (C_5 + C_6))$, and we choose $\epsilon = \frac{c\gamma}{\xi}$, and combine the above results. Then at least with probability $1 - O(N_\epsilon \exp(-n\frac{c^2 s \gamma^2}{\xi^2}))$, we have $\|\nabla R(\boldsymbol{w}^{(2)})\|_2 > \frac{\gamma}{3}$. Therefore we can choose $n \gtrsim \frac{\xi^2}{c^2\gamma^2}\log(\frac{((\sqrt{s_0}+\tau\sqrt{p})^2+\sqrt{p}t)C_1 C_2 C_5 C_6 C_8 C_x}{c^3\gamma})(pd_v d + d + 2pd_q)$, such then $N_\epsilon \exp(-n\frac{c^2\gamma^2}{\xi^2}) = o(1)$. Thus with this required sample complexity, we have $\|\nabla R(\boldsymbol{w}^{(2)}) - \mathbb{E}(\nabla R(\boldsymbol{w}^{(2)})\|_2 \leq \frac{2c\gamma}{3}$.

Finally we conclude that with high probability, any parameter $(\boldsymbol{w}^{(1)}, \boldsymbol{w}^{(2)}, \boldsymbol{w}^{(v)}, \boldsymbol{w}^{(k)}, \boldsymbol{w}^{(q)})$ with $\mathbb{E}(u^2) \geq \gamma$, we have $\|\nabla R(\boldsymbol{w}^{(2)})\|_2 > \frac{\gamma}{3}$. Then following the same empirical risk convergence argument as Lemma 1.2, with high probability they cannot be stationary point. We conclude the sample complexity bound as:

$$n \gtrsim \frac{(\sqrt{s_0} + \tau\sqrt{p})^2 + \sqrt{p}t)^2 pd_v d\eta_a^2}{\gamma^2}\log(\frac{((\sqrt{s_0} + \tau\sqrt{p})^2 + \sqrt{p}t)\eta_a \eta_b}{\gamma})$$

$\square$

### E.5  PROOF OF THEOREM 3

*Proof.* Under assumption (A3.1), we know all input features and weights are bounded. Therefore we know $\|\nabla f(\boldsymbol{w}^{k_r})\|_2$ is a Lipschitz continuous function on all parameters, and we denote its Lipschitz constant $c_k$. For $\boldsymbol{w}^{k_r}$, we can derive that:

$$\nabla R(\boldsymbol{w}^{k_r}) = \frac{1}{n}\sum_{i=1}^{n} u_i \nabla h(\boldsymbol{w}^{k_r})$$

Under (A6), if we have $\mathbb{E}(h(\boldsymbol{x}) - \mathbb{E}(y|\boldsymbol{x}))^2 \gtrsim \gamma^2$, then:

$$\|E(\nabla R(\boldsymbol{w}^{k_r}))\|_2 = \|cov(\nabla h(\boldsymbol{w}^{k_r}), \boldsymbol{u})\|_2 \gtrsim O(\gamma)$$

Then similar with Theorem 1 and 2, we construct an $\epsilon$-cover over all parameters $\boldsymbol{\theta} := (\boldsymbol{w}^{k_1}, \boldsymbol{w}^{k_2}, \boldsymbol{w}^V, \boldsymbol{w}^Q, \boldsymbol{w}^K)$, and we denote it as $\{\boldsymbol{\theta}_1, \ldots, \boldsymbol{\theta}_N\}$ such that for any feasible parameter, there exist $j \in [N]$ such that the maximum $\ell_2$ distance to $\boldsymbol{\theta}_j$ is smaller than $\epsilon$. By calculating the number of parameters in all matrices in $\boldsymbol{\theta}$, we have

$$\log(N_\epsilon) = \frac{1}{\epsilon}O(d_{self} + \sum_{i=1}^{k}(d_k + d_v)q_k)$$

Denoting $\nabla R(\boldsymbol{w}_{(j)}^{k_r})$ as the gradient with respect to $j^{th}$ parameter set in $\epsilon$-cover for $j \in \{1, \ldots, N_\epsilon\}$:

$$P(\|\nabla R(\boldsymbol{w}_{(j)}^{k_r}) - \mathbb{E}_x(\nabla R(\boldsymbol{w}_{(j)}^{k_r})\|_2 \geq \frac{c\gamma}{3}) \lesssim \exp(-n\frac{c^2\gamma^2}{c_k^2})$$

By union bound, we have:

$$P(\exists j \in [N_\epsilon], \|\nabla R(\boldsymbol{w}_{(j)}^{(2)})\|_2 \geq \frac{2c\gamma}{3}) \lesssim N_\epsilon \exp(-n\frac{c^2\gamma^2}{c_k^2})$$

Secondly we analyze $\|\nabla R(\boldsymbol{w}^{k_r}) - \nabla R(\boldsymbol{w}_{(j)}^{k_r})\|_2$ term. As we have shown that the gradient is Lipschitz continuous, thus we have:

$$\|\nabla R(\boldsymbol{w}^{k_r}) - \nabla R(\boldsymbol{w}_{(j)}^{k_r})\|_2 \leq c_K \epsilon$$

We choose $\epsilon = \frac{c\gamma}{3c_K}$, then at least with probability $1 - \mathcal{O}(N_\epsilon \exp(-n\frac{c^2\gamma^2}{c_K}))$, we have $\|\nabla R(\boldsymbol{w}^{(2)})\|_2 > \frac{c\gamma}{3}$. Therefore we can choose $n \gtrsim \log(\frac{c_k}{\gamma})(d_{self} + D(d_k + d_v q_k))$, such that $N_\epsilon \exp(-n\frac{c^2\gamma^2}{c_k^2}) = o(1)$. Finally we can conclude that with probability $1 - o_n(1)$, for any $(\boldsymbol{a}, \boldsymbol{w}^{(1)}, \boldsymbol{w}^{(2)})$ such that $\mathbb{E}(\tilde{\boldsymbol{w}}^{(2)}\phi(\langle \tilde{\boldsymbol{w}}^{(1)}, \boldsymbol{x} \odot \tilde{\boldsymbol{a}}\rangle) - \mathbb{E}(y|\boldsymbol{x}))^2 \geq \gamma$, we have $\|\nabla R(\boldsymbol{w}^{(2)})\|_2 > \frac{c\gamma}{3}$. Then following the convergence of empirical risk procedure of Lemma 1.2, we show with probability going to 1 such that $\|\nabla R_n(\boldsymbol{w}^{(2)})\|_2 > 0$ and all parameters with prediction error $O(\gamma)$ cannot be stationary point as long as $n \gtrsim \log(\frac{c_k}{\gamma})(d_{self} + D(d_k + d_v q_k))$. Thus we complete the proof. □

### E.6 PROOF OF THEOREM 4

*Proof.* First with $\|\boldsymbol{f}(\boldsymbol{x})\|_0 = s_0$, we know all the inputs $\boldsymbol{x}_i$ with corresponding $\boldsymbol{a}_i = 0$, will be inactive in the network. We can omit all these inactive inputs. Then we split $n_1$ units into $s_0$ group, with $\lfloor \frac{n_1}{s_0} \rfloor$ number of units in each group, and discard the leftover units. $s_0$ different groups correspond to $s_0$ active inputs with non-zero attention weight.

Inside each group, for example in $j^{th}$ group, denoting $q = \lfloor \frac{n_1}{s_0} \rfloor$, we choose the input weights and biases for $i = 1, 2, \cdots, q$ as:

$$h_1(\boldsymbol{x}) = \max\{0, \boldsymbol{w}_j \boldsymbol{x}\},$$
$$h_2(\boldsymbol{x}) = \max\{0, 2\boldsymbol{w}_j \boldsymbol{x} - 1\},$$
$$\vdots$$
$$h_q(\boldsymbol{x}) = \max\{0, 2\boldsymbol{w}_j \boldsymbol{x} - (q-1)\}$$

here we assign $\boldsymbol{w}_j$ to be a row vector with $j^{th}$ variable equal to 1 and all other entries to be 0. And in the second layer, we choose $\boldsymbol{w}^{(2)} = (\boldsymbol{w}_3, \cdots, \boldsymbol{w}_3)$, where $\boldsymbol{w}_3 = (1, -1, 1, \cdots, (-1)^{q+1})$, corresponding to $h_1$ to $h_q$ in each group. Then the designed network has $q$ linear regions inside each group, giving by the intervals:

$$(-\infty, 0], (0, 1], (1, 2], \cdots, [q-1, \infty)$$

Each of these intervals has a subset that is mapped by $\boldsymbol{w}_3 h(\boldsymbol{x})$ onto the interval (0,1).Montufar et al. (2014) Therefore the total number of linear regions is lower bounded by $\lfloor \frac{n_1}{s_0} \rfloor^{s_0}$. □

### E.7 PROOF OF THEOREM 5

*Proof.* Here we define an $\alpha$ scale transformation such that:

$$T_\alpha : (\boldsymbol{w}^{(1)}, \boldsymbol{w}^{(2)}) \mapsto (\alpha \boldsymbol{w}^{(1)}, \alpha^{-1}\boldsymbol{w}^{(2)})$$

And all the value,query and key matrices remain the same. Then we know the Jacobian determinant for $T_\alpha = \alpha^{(pd_v-1)d}$. Since $pd_v d \geq d$, as we assign $\alpha \to \infty$, such that the Jacobian determinant goes to infinity, and the volume of $C(L, \boldsymbol{\theta}, \epsilon)$ goes to infinity.

For the Hessian matrix, we still assume a positive diagonal element $\delta > 0$ in $\boldsymbol{w}^{(1)}$. Similarly we have the Frobenius norm $\|\nabla^2 L(T_\alpha(\boldsymbol{\theta}))\|_F$ of

$$\nabla^2 L(T_\alpha(\boldsymbol{\theta})) = \begin{bmatrix} \alpha^{-1}\boldsymbol{I} & 0 \\ 0 & \alpha\boldsymbol{I} \end{bmatrix} \nabla^2 L(\boldsymbol{\theta}) \begin{bmatrix} \alpha^{-1}\boldsymbol{I} & 0 \\ 0 & \alpha\boldsymbol{I} \end{bmatrix}$$

is lower bounded by $\alpha^{-2}\delta$. When we choose sufficient small $\alpha$, we have the biggest eigenvalue of $\nabla^2 L(T_{\alpha_1, \alpha_2}(\boldsymbol{\theta}))$ is larger than any constant $M$. Therefore there exists a stationary point such that the operator norm for Hessian is arbitrary large. □

### E.8 PROOF OF THEOREM 6

*Proof.* First, we obtained new $\epsilon$-covering bound for the parameter set $(\boldsymbol{w}^{(2)}, \boldsymbol{w}^{(1)}, \boldsymbol{w}_f, \boldsymbol{w}_a)$:

$$N_{\epsilon_1} \le (3C_1/\epsilon)^d, N_{\epsilon_2} \le (3C_2/\epsilon)^{td}, N_{\epsilon_3} \le (3C_8/\epsilon)^{d_a}, N_{\epsilon_4} \le (3C_9/\epsilon)^{d_f},$$

And $N_\epsilon \le \Pi_{i=1}^4 N_{\epsilon_i}$ Similar to Theorem 1, we denote

$$u = (\boldsymbol{w}^{(2)T}\phi(\langle\boldsymbol{w}^{(1)}, \sum_{j=1}^p \boldsymbol{a}(\boldsymbol{x}_i)\boldsymbol{x}_i^j) - \mathbb{E}(y|\boldsymbol{x})) \tag{15}$$

Then the derivatives of population risk with expectation to $\boldsymbol{y}$ can be presented as follows:

$$\nabla R(\boldsymbol{w}^{(2)}) = \frac{1}{n}\sum_{i=1}^n u_i\phi(\langle\boldsymbol{w}^{(1)}, \sum_{j=1}^p \boldsymbol{a}(\boldsymbol{x}_i)\boldsymbol{x}_i^j)$$

And also, the new $\boldsymbol{v}_i$ term is: $\boldsymbol{v}_i = \boldsymbol{u}_i\phi(\langle\boldsymbol{w}_j^{(1)}, \sum_{j=1}^p \boldsymbol{a}(\boldsymbol{x}_i)\boldsymbol{x}_i^j\rangle)$. Following (A15) and same procedure as Lemma 1.1, we have $\|\mathbb{E}(\nabla R(\boldsymbol{w}^{(2)}))\|_2 \ge c\gamma$. we have new bound of $\|\boldsymbol{v}\|_2^2$ with respect to $\boldsymbol{x}$ is upper bounded by $\sigma^2 = t^2C_1^4C_2^2C_8^2C_9^2C_X^4$ with normalized attention weight. Same argument follows for the case when $\|\mathbb{E}(\nabla R(\boldsymbol{w}^{(1)}))_k\|_2 \ge c\gamma$. Then following the same approach as Theorem 1 and 2, we obtain the sample complexity bound:

$$n \gtrsim \frac{\sigma^2}{c^2\gamma^2}\log(\frac{tC_1C_2C_8C_9C_x}{c^3\gamma})(d + td + d_f + d_a)$$

$\square$

### E.9 PROOF OF THEOREM 7

*Proof.* Here we only consider the derivatives with respect to $\boldsymbol{w}^{(1)}$ and $\boldsymbol{w}^{(2)}$, they can be presented as follows:

$$\nabla R_n(\boldsymbol{w}^{(2)}) = \frac{1}{n}\sum_{i=1}^n u_i\phi(\langle\boldsymbol{w}^{(1)}, vec(\boldsymbol{w}^V\boldsymbol{x}_i\boldsymbol{a}_i^{self})\rangle)$$

$$\nabla R_n(\boldsymbol{w}^{(1)}) = \frac{1}{n}\sum_{i=1}^n u_i(vec(\boldsymbol{w}^V\boldsymbol{x}_i\boldsymbol{a}_i^{self}))(\boldsymbol{w}^{(2)} \odot \phi'(\langle\boldsymbol{w}^{(1)}, vec(\boldsymbol{w}^V\boldsymbol{x}_i\boldsymbol{a}_i^{self})\rangle))^T$$

By assumption, $rank(\phi(\langle\boldsymbol{w}^{(1)}, vec(\boldsymbol{w}^V\boldsymbol{x}_i\boldsymbol{a}_i^{self})\rangle))_{i=1,...,n}) = n$. Solving the linear system, we must have $u_i = 0$ for any $i = 1, 2, ..., n$ to satisfy that $\nabla R_n(\tilde{\boldsymbol{w}}^{(2)}) = 0$. Thus we know that the loss is exactly zero inside sample. Thus it must be a global minimum. $\square$

## F FIXED ATTENTION MASK EXPERIMENTS ON NOISYMNIST

In this additional experiments, another variant of NoisyMNIST dataset is constructed where the images of digits from the MNIST dataset are embedded in noise, as shown in Figure (a). Then we compare a two-layer MLP baseline model (in blue) with a global (fixed) attention model proposed in Supplemental D (in red). Fig (b) shows a plot of epochs v.s. test loss while Fig (c) shows a plot of epochs v.s. test accuracy. We observe that the global attention model obtains lower loss values and higher test accuracy with fewer epochs. Again it verifies the superiority of attention weights are from the effect of concentrating attention weights.

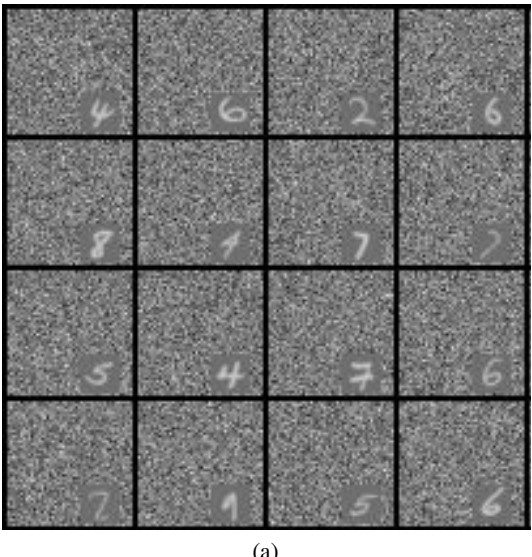

(a)

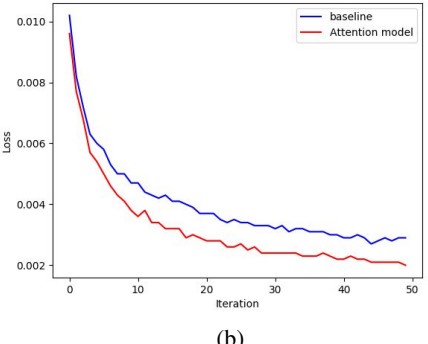

(b)

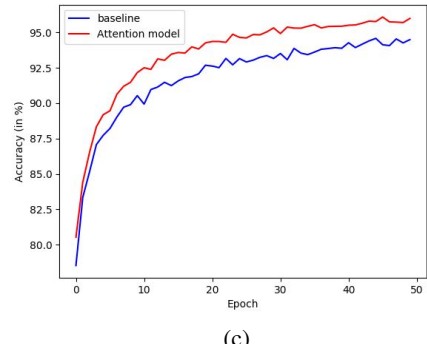

(c)

