# OpenReview forum: "Analyzing Attention Mechanisms through Lens of Sample Complexity and Loss Landscape"
_ICLR.cc/2021/Conference — Reject_

### Official Review · AnonReviewer2 · 2020-10-26
**Interesting idea but with too strong assumptions and some black boxes unopened**

**Rating:** 5
**Confidence:** 3

**Review:**

Summary:

This paper analyzes the sample complexity and properties of loss landscape for self-attention neural networks. Specifically, under the assumptions about teacher-student setting, "sparsity" of the attention mask, enough over-parametrized student model, and regularity assumptions (all the weights are bounded, etc.), in a regression task with square loss, the authors proved that a two-layer MLP with attention enjoys a smaller upper bound on the sample complexity compare to the baseline without attention. Using very similar techniques and proof procedures with stronger assumptions, this result can be extended to multi-layer or recurrent attention models. Experiments were done on NLP and CV tasks with shallow networks/small datasets to validate the theoretical results.

Main assumptions:

The assumptions for different theorems are slightly different, but they are somewhat parallel and can be summarized into the following categories. Each theorem requires one assumption from each category.

1. Teacher-student setting: There exists a two-layer teacher model with attention that the ground-truth labels are the outputs of the teacher network plus i.i.d. sub-gaussian noise.

2. The largest few (less than input dimension) weights in attention mask has a constant share of the norm of the entire mask, i.e., the weights of the attention mask are sparse/highly concentrated.

3. When the sample size is large enough (tend to infinity), the student model with learned weights is significantly close to the teacher model in terms of covariance matrix of residual w.r.t. linear regression.

4. Regularity assumptions, e.g., the norms of all the weights and samples are bounded.

Main results:

1. For two-layer neural networks, under all the assumptions mentioned above, the authors proved that the model with a fixed attention mask has a smaller sample complexity upper bound compare to that without an attention mask. By using the same procedure and techniques with stronger assumptions, the result can be extended to attention with a normal self-attention mask, multi-layer self-attention models, and recurrent attention models. The proofs basically come from the same lemma (Lemma 1 in appendix) and can be considered as different corollaries of Lemma 1 under different sets of assumptions.

2. Experiments were done to validate the theoretical results. Specifically, the authors did experiments on Portuguese to English translation, IMDB reviews sentiment classification, and noisy MNIST with small models, e.g., two-layer conv nets or fully-connected nets with/without attention. The experiments showed that the weights of the attention mask are concentrated, and attention in these tasks has the effect of reducing sample complexity and improving performance.

3. The authors also showed that the sparse attention model has a smaller lower bound on the maximal number of linear regions and suffers from the scaling issue when considering the flatness of minima.

Pros:

1. The problem analyzed in this paper, i.e., understanding the underlying reasons why attention works, is an important problem, and this paper provides some theoretical together with experimental evidence for this.

2. This paper is written in a clear and well-organized way so it is easy to follow. The experimental settings are explained in detail, and the experiments are replicated multiple times.

3. The theoretical proof and experimental methods appear to be correct and reasonable.

Cons:

1. The authors seem to use attention somewhat like a black box. The weight matrices, e.g., queries, keys, values, are learned during the training process, but the authors just assume that they have some nice properties as stated in Assumption A3. This makes the attention mechanism in the proof essentially the same as the known fixed function f. However, I think one of the core parts of the attention mechanism and part of the reason why it works is the learning process of the query, key, value matrices, which is ignored by the authors in this paper.

2. Similar to 1, the authors also hide other properties of the loss landscape in assumption A3. This assumption is not only about over-parametrization because it is a universal property of all possible weights along the training trajectory or on the entire landscape. I believe that over-parametrization can be shown to prove the existence of such weights, but this assumption is much stronger than the over-parametrization itself because another important thing is how the training algorithms find such points. This is also part of the reason why the authors can have strong results that hold for *any stationary point*.

3. The comparisons in this paper are about the upper bounds or lower bounds, but there is a lack of evidence about the tightness of these bounds. For instance, for the comparison between the sample complexity for models with/without attention, the bounds seem to be very large and a bit loose since empirically the networks can achieve small error with much smaller sample sizes. Therefore, it is somewhat unfair to do a direct comparison between the upper bounds. Besides, the loss landscape of models with and without attention may have different $\gamma$, which is not addressed by the authors in the paper.

4. The logic for this flatness part is kind of confusing to me. The scaling issue is to show that flatness alone cannot fully explain the generalization performance, but not having that issue does not mean the model can generalize better. Besides, the bound on the model weights are the assumptions, not conclusions, of this paper. The authors did not rigorously prove that their model does not suffer from scaling issues.

5. For the noisy-MNIST experiments, the authors were claiming that effect of attention is not mainly due to the model structure by showing that an equal weight attention model performs poorly. However, the authors did not decouple the performance gain achieved by attention in the concentration of the weights and the learning process of the mask. I wonder whether the model will perform well if a fixed concentrated instead of equal weight mask is applied. If the concentration of the mask weights is very important, it should improve the performance of the model by a lot.

Recommendation:

I tend to vote for rejecting this paper. My biggest concern about this paper is that this paper considered some parts of the problem as black boxes and directly have some assumption about them (in other words, the assumptions made by the authors, especially A3, are too strong and hide a lot of things like implicit dependencies on network width, attention mechanism and model structure in parameter $\gamma$), but some of the things skipped by the authors may be the core part of the problem, making the analysis provided in this paper a bit superficial and the results somewhat limited.

Supporting arguments for recommendation:

See Cons, especially points 1 and 2 there.

Questions for the authors:

1. Please address the cons mentioned above.

Additional feedback:

There are some notation confusions. For instance, $\tau$ is used as both the parameter for the softplus activation function (last paragraph of page 2) and in assumption A2. Besides, for the paragraph before assumption (A1) and assumptions (A1) - (A3), the authors are using both $x\odot f(x)$ and $x_i\odot f(x)$.

---

> ### Author Response · Authors · 2020-11-13
> **Detailed response to Reviewer 2 part 1**
>
> Thank you very much for your valuable comments. In what follows, we repeat your comments and explain how we addressed them.
>
> Cons 1: The authors seem to use attention somewhat like a black box. The weight matrices, e.g., queries, keys, values, are learned during the training process, but the authors just assume that they have some nice properties as stated in Assumption A3. This makes the attention mechanism in the proof essentially the same as the known fixed function f. However, I think one of the core parts of the attention mechanism and part of the reason why it works is the learning process of the query, key, value matrices, which is ignored by the authors in this paper.
>
> Answer: We agree that our analysis does not heavily depend on the weight matrices or other blocks defined as ‘black-box’ by the reviewer. Reasons are as follows: (1) We think the most important part of attention models is the mechanisms of determining the attention weights. Such attention weights tell the model how relevant between features and how to proceed. Our analyses observe that the concentration of attention weights is how attention models can constrain the parameter space and reduce the unnecessary variability in gradients. We think this is a crucial part of attention models, and our theory heavily emphasizes this part. (2) The ‘black-box’ part mentioned by the reviewer, actually is crucial in providing the strong expressiveness of the model. Yang et al.(2019) prove the strong expressiveness of transformer model for any sequence-to-sequence function. Such results heavily rely on the key, query, weight structure of the model. Since our paper focuses on the sample complexity and convergence rate of attention model, assuming that such model provides us strong enough expressiveness. That’s why our results are not heavily dependent on these blocks of the model.
>
> Cons 2: Similar to 1, the authors also hide other properties of the loss landscape in assumption A3. This assumption is not only about over-parametrization because it is a universal property of all possible weights along the training trajectory or on the entire landscape. I believe that over-parametrization can be shown to prove the existence of such weights, but this assumption is much stronger than the over-parametrization itself because another important thing is how the training algorithms find such points. This is also part of the reason why the authors can have strong results that hold for any stationary point.
> Answer: In Appendix A.1 page 12, we provide the experiments to verify this assumption. We generate a two-layer network under self-attention model (4) with $n=500$, $d=256$, $p=2142$, $d_v=100$. We choose the set of parameter same with our experiment setup in Section 6, and hidden layers are sufficient overparameterized with $p=2142$. Then we compute the empirical conditional covariance $\hat{\lambda}_{\max}(Var(\phi(\langle w^{(1)\star}, x_i\odot f(x)\rangle)|\phi(\langle w^{(1)}, x\odot f(x)\rangle))=3.13*10^{-8}$. Therefore this term is very small, and it holds almost everywhere. Therefore we can safely assume that in the training trajectory we will not encounter points such that violating this assumption.
>
> Cons 3: The comparisons in this paper are about the upper bounds or lower bounds, but there is a lack of evidence about the tightness of these bounds. For instance, for the comparison between the sample complexity for models with/without attention, the bounds seem to be very large and a bit loose since empirically the networks can achieve small error with much smaller sample sizes. Therefore, it is somewhat unfair to do a direct comparison between the upper bounds. Besides, the loss landscape of models with and without attention may have different $\gamma$, which is not addressed by the authors in the paper.
>
> Answer: The upper bound on sample complexity is tight in the aspect of comparison. Appendix E pages 18-21 provides details on how the bounds are derived. In derivations, the inequalities all can take inequalities. We summarize them as follows:(1) For the concentration inequality, we use Cauchy-Schwarz inequality which is tight when $w^{(1)}$ and $x\cdots a$ are in the same direction and $x \cdots a$ is non-sparse. We apply Hoeffeding bound at the beginning of page 20, which is also a tight inequality. (2) In Lemma 1.3 on page 20, The derivation on central limit theorem is also tight in order. (3) On page 21, after separating the leading and non-leading attention weights, we apply Cauchy-Schwarz inequality, in which inequality can take equal if the weights and attention are in the same direction. Therefore, we are comparing two tight bounds of sample complexity.

---

> > ### Author Response · Authors · 2020-11-13
> > **Detailed response to review 2 part 2**
> >
> > Cons 4: The logic for this flatness part is kind of confusing to me. The scaling issue is to show that flatness alone cannot fully explain the generalization performance, but not having that issue does not mean the model can generalize better. Besides, the bound on the model weights are the assumptions, not conclusions, of this paper. The authors did not rigorously prove that their model does not suffer from scaling issues.
> >
> > Answer: Also from the sample complexity bound of Theorem 1, up to a log term, prediction error $\gamma$ is proportional to $n^{-1/2}$, which is the optimal rate of convergence in regression. In Imaizumi and Fukumizu(2018) and similar works, they showed the generalization error convergence rate is $O(n^{-t})$ where $0<t<1/2$. These facts all imply that the bound is tight in the aspect of sample size compared with existing works.
> >
> > Therefore in the aspect of tightness, we think the sample complexity from attention model and the baseline are comparable.
> >
> > For $\gamma$ in (A3), as we justified theoretically and empirically, the residual can be as small as to $10^{-8}$, and this part does not affect the sample complexity bound as long as it is smaller than the given $\gamma$. Therefore it will not hurt the fairness of comparison.
> >
> > The scaling issue. In math, as long as we impose $\ell_1$ and $\ell_2$ regularization, it is equivalent to impose some $\ell_1$ and $\ell_2$ bound on weights. Thinking about a model with infinity weights, it can never achieve minimal loss given such regularization. Therefore our assumption can be achieved as long as we provide regularizations. In this part, we don’t argue why attention model is superior, but arguing that attention models will not hurt the flatness property of the non-attention model, given the same bound on network weights.
> >
> > Cons 5: For the noisy-MNIST experiments, the authors were claiming that effect of attention is not mainly due to the model structure by showing that an equal weight attention model performs poorly. However, the authors did not decouple the performance gain achieved by attention in the concentration of the weights and the learning process of the mask. I wonder whether the model will perform well if a fixed concentrated instead of equal weight mask is applied. If the concentration of the mask weights is very important, it should improve the performance of the model by a lot.
> >
> > Answer: In our noisy MNIST experiments, fixed concentrated maps are not available because the true locations of digits are different for each image. In the revised paper Appendix F, we conduct an additional experiment with a fixed location on each image and learn a fixed attention mask for all images. The result shows that the fixed attention mask model performs much better than the baseline. Thus it verifies the superiority of attention weights are from the effect of concentrating attention weights.
> >
> > At last, thanks so much for pointing out the typos in our paper. We have updated them accordingly.

---

### Official Review · AnonReviewer3 · 2020-10-28
**ICLR 2021 Conference Paper2164 AnonReviewer3**

**Rating:** 3
**Confidence:** 3

**Review:**

Summary:
This paper studies attention mechanisms.The main results of the paper are upper bounds on sample complexities of attention models and the baseline non-attention model. The paper also further investigates several additional properties and provides empirical studies.

Concerns:
(1) Without going into the details, the main problem of the paper is that it only gives upper bounds on the sample complexity. When comparing the upper bound of two algorithms, the comparison does not show which one is better. To show one algorithm is better than the other, you need to prove that one’s upper bound is smaller than the other’s lower bound on sample complexity.
(2) The paper proves the result only for a specific non-attention model. Without further justification, it is very unclear why this is enough to represent other non-attention models.

Reasons for score:
I vote for rejection. The theoretical results are not sufficient to support the claims made in the paper about attention models.

---

> ### Author Response · Authors · 2020-11-13
> **Detailed response to Reviewer 3**
>
> Thank you very much for your valuable comments. In what follows, we repeat your comments and explain how we addressed them.
>
> Concern 1: Without going into the details, the main problem of the paper is that it only gives upper bounds on the sample complexity. When comparing the upper bound of two algorithms, the comparison does not show which one is better. To show one algorithm is better than the other, you need to prove that one’s upper bound is smaller than the other’s lower bound on sample complexity.
>
> Answer: The concern on the comparison of bounds: The upper bound on sample complexity is tight in the aspect of comparison. Appendix E pages 18-21 provides details on how the bounds are derived. In derivations, the inequalities all can take inequalities. We summarize them as follows:(1) For the concentration inequality, we use Cauchy-Schwarz inequality which is tight when $w^{(1)}$ and $x\cdots a$ are in the same direction and $x \cdots a$ are non-sparse. We apply Hoeffeding bound at the beginning of page 20, which is also a tight inequality. (2) In Lemma 1.3 on page 20, The derivation on central limit theorem is also tight in order. (3) On page 21, after separating the leading and non-leading attention weights, we apply Cauchy-Schwarz inequality, in which inequality can take equal if the weights and attention are in the same direction. Therefore, we are comparing two tight bounds of sample complexity.
>
> Also from the sample complexity bound of Theorem 1, up to a log term, prediction error $\gamma$ is proportional to $n^{-1/2}$, which is the optimal rate of convergence in regression. In Imaizumi and Fukumizu(2018) and similar works, they showed the generalization error convergence rate is $O(n^{-t})$ where $0<t<1/2$. These facts all imply that the bound is tight in the aspect of sample size compared with existing works.
>
> Therefore in the aspect of tightness, we think the sample complexity from attention model and the baseline are comparable.
>
> Concern 2: The paper proves the result only for a specific non-attention model. Without further justification, it is very unclear why this is enough to represent other non-attention models.
>
> Answer: Our contribution includes several parts. The first part is the comparison of attention model vs. non-attention baseline models. The second part of contribution is analyzing the transformer type self-attention models(Theorem 3). Our analyses show that self-attention model can reduce sample complexity by properly designing concentrated attention weights. We also provide guidance on attention model designing in practice in Section 5. The third part of contribution is analyzing the recurrent attention models(Appendix Section B). Both the second and third part is one step forward understanding the mechanisms of attention models and provide practical guidance in model training.

---

### Official Review · AnonReviewer4 · 2020-10-30
**Strong assumptions**

**Rating:** 4
**Confidence:** 4

**Review:**

# Summary

The paper presents a theoretical analysis of self-attention mechanisms in neural networks by considering a two layer neural network with known and learned attention mechanisms. The paper shows that in the non-agnostic case, i.e. assuming that data is generated from a function in the function class of two-layered neural networks with self-attention, then every local minima of the squared loss has low expected generalization error. The theoretical analysis reveals that using neural networks with self-attention can have lower sample complexity than standard 2-layered neural networks when the data is in fact generated by a two-layered neural network with concentrated self-attention weights. The paper justifies some of the assumptions through experiments on IMDB reviews data set and also shows both theoretically and empirically that weight regularization can improve generalization.

# Strength

1. While attention mechanisms have had tremendous empirical success in NLP, little is known about their theoretical properties. Towards that end the paper takes a first stab at theoretically analyzing attention mechanisms by considering the expected generalization error of local minima of the loss functions.
2. Through some toy experiments the paper reveals that weight regularization can improve generalization of two-layered NNs with self attention.

# Weakness

1. The paper goes for too many results at the cost of readability. Lots of results (for recurrent self-attention) are deferred to Appendix and even formal statements of Theorem are not in the main paper.
2. While analyzing two-layered NNs is acceptable for making progress, the main problem is the non-agnostic setting where the authors assume that data is being generated by a two-layered NN with concentrated self-attention. Existing results for two-layered NNs (e.g. Allen-Zhu et al. 2017) are for the agnostic case. This is a significant shortcoming of the current work.
3. Better sample complexity of attention model comes from the assumption that the true generative model has concentrated self-attention weights, i.e. most of the attention weights are concentrated in a few attention masks. So fitting a model that exploits this assumption, , i.e. a model that belongs to the same class as the true function, would have better sample complexity.
4. Apart from the non-agnostic assumption, the paper also makes a number of very unrealistic assumptions on the weights (Assumption A3, A1.3 and A4) of the model class. All these assumptions make assumptions on not only the true weights (the weights of the NN generating data) but on the weights of all NNs in the model class. So it is not clear if the model class is even realizable.
5. There are also many problems with experiments which are detailed below.

# Questions to address in rebuttal

1. For which w's does Assumption A3 (maximum eigenvalue of covariance matrix of residual), A1.3 (zero mean), and A4 (query, key, value) need to hold? If this needs to hold for all w's (and not just the true w's) then this is a very strong assumption. Is the model class realizable?
2. "Assumption A1 (non-agnostic setting) is supported by previous works" which works? Needs reference. Experiment 1 shows that the learned self attention weights are concentrated not that the true data is being generated from such a model.
3. What is k in Assumption A3 ?
4. Seems like $1/\sqrt{n}$ rate comes from the assumption that maximum eigenvalue is $o(\gamma^2)$ for some $\gamma > 0$ ? Why use $\gamma^2$ ? Why not just $\gamma ?$
5. Not clear where assumption A3 is used in the proof. Apparently A3 is used in proof of Theorem 3 but A3 is also needed in proof of Theorem 1. It is stated in proof of Theorem 3 that "by Assumption A3.1 we know all input features and weights are bounded." but that is not what A3 states.
6. The theoretical claim is that l2 norm of weights determines generalization error but experiments use l1 norm. Why?
7. Analysis is for regression tasks but experiments are on classification.
8. Why noisy MNIST ? Why not on the original MNIST ? It is obvious that paying attention on the digit blocks and ignoring the uniform noise blocks will help.

# Minor comments

1. we first introduced a two-layer non-attention model ⇒ we first introduce a two-layer non-attention model.
2. we consider attention weights are determined ⇒ we consider attention weights that are determined.
3. There are also a number of other grammatical mistakes that should be fixed to improve readability.

---

> ### Author Response · Authors · 2020-11-13
> **Detailed response to Reviewer 4 part 1**
>
> Thank you very much for your valuable comments. In what follows, we repeat your comments and explain how we addressed them.
> Weakness 1: The paper goes for too many results at the cost of readability. Lots of results (for recurrent self-attention) are deferred to Appendix and even formal statements of Theorem are not in the main paper.
>
> Answer: Due to the page limit, we tried to summarize the assumptions intuitively to improve the readability. We tried our best to summarize the high-level idea of assumptions in the main paper. Sorry for the confusion caused by this.
>
> Weakness 2: While analyzing two-layered NNs is acceptable for making progress, the main problem is the non-agnostic setting where the authors assume that data is being generated by a two-layered NN with concentrated self-attention. Existing results for two-layered NNs (e.g. Allen-Zhu et al. 2017) are for the agnostic case. This is a significant shortcoming of the current work.
>
> Answer: We analyze a two-layer network since we believe it can convey the main discovery and ideas of our paper: Why attention model improves the performance, and how to design attention models making the best use out of it. Beyond two-layer network, We also study the multi-layer neural nets in Section 3.3 Theorem 3 and recurrent neural nets in Appendix Section B.
>
> Weakness 3: Better sample complexity of attention model comes from the assumption that the true generative model has concentrated self-attention weights, i.e. most of the attention weights are concentrated in a few attention masks. So fitting a model that exploits this assumption, , i.e. a model that belongs to the same class as the true function, would have better sample complexity.
>
> Answer: We think generative assumption (A1) is not a strong assumption, considering the massive literature proves the strong expressiveness of neural nets. Specifically for two-layer neural nets and transformer model, Barron & Klusowski 2018 shows that general bounded functions with a Fourier representation on [-1,1] can be well approximated by the defined two-layer network. For the transformer model, Yun et al.(2019) prove the strong expressiveness of transformer type self-attention model for any bounded sequence-to-sequence functions.
>
> Weakness 4: Apart from the non-agnostic assumption, the paper also makes a number of very unrealistic assumptions on the weights (Assumption A3, A1.3, and A4) of the model class. All these assumptions make assumptions on not only the true weights (the weights of the NN generating data) but on the weights of all NNs in the model class. So it is not clear if the model class is even realizable.
>
> Answer: We assume weights have a specific $\ell_1$ and $\ell_2$ bound. In math, as long as we impose $\ell_1$ and $\ell_2$ regularization, it is equivalent to impose some $\ell_1$ and $\ell_2$ bound on weights. Thinking about a model with infinity weights, it can never achieve minimal loss given such regularization. Therefore such assumptions are practical, and further emphasize the importance of regularizations of these weights, as we suggest in Section 5.
>
> Question 1: For which w's does Assumption A3 (maximum eigenvalue of covariance matrix of residual), A1.3 (zero mean), and A4 (query, key, value) need to hold? If this needs to hold for all w's (and not just the true w's) then this is a very strong assumption. Is the model class realizable?
>
> Answer: We only require it holds for w1 and w2, but not weights for the query, key, value matrices.
>
> Question 2: "Assumption A1 (non-agnostic setting) is supported by previous works" which works? Needs reference. Experiment 1 shows that the learned self-attention weights are concentrated not that the true data is being generated from such a model.
>
> Answer: In assumption (A1), we assume the two-layer network has sufficiently strong expressiveness of the true generative model. As we have discussed, Barron & Klusowski 2018 shows that general bounded functions with a Fourier representation on [-1,1] can be well approximated by the defined two-layer network. For transformer model, Yun et al.(2019) prove the strong expressiveness of transformer type self-attention model for any bounded sequence-to-sequence functions.

---

> > ### Author Response · Authors · 2020-11-13
> > **Detailed response to Reviewer 4 part 2**
> >
> > Question 3: What is k in Assumption A3 ?
> >
> > Answer: Thanks for pointing out that we missed the interpretation of k. Actually, k is a constant satisfying $k<\sqrt{s_0}+\tau\sqrt{p}$, i.e. given the sample complexity bound, the hidden layers are sufficiently overparameterized such that the linear combinations of them can explain most of the true hidden units. We justify this assumption both theoretically and empirically. Theoretically, we show that when $d$ is large, given the $2^d$ linear regions and strong expressiveness of $\{\phi_{1\dots d}(\langle w^{(1)},x\odot f(x)\rangle)\}$, the residual vector is a $o(\gamma)$ term with respect to all directions in $\mathbb{R}^d$. Also, it coincides with the assumption in Nguyen et al. 2017 and the theoretical result in Allen-Zhu et al. 2019. Empirically, we also evaluate the residual term in a two-layer network with self-attention model. It shows the largest eigenvalue of the residual covariance matrix is as small as $3.13*10^{-8}$. Therefore we can assume it to be very small. The detailed justifications are provided in Appendix A.1 on page 11.
> >
> > Question 4: Seems like $1/\sqrt{n}$ rate comes from the assumption that maximum eigenvalue is $o(\gamma^2)$ for some $gamma > 0$? Why use $\gamma^2$? Why not just $\gamma$?
> >
> > Answer: $1/\sqrt{n}$ rate does not come from the $o(\gamma^2)$, actually it comes from the concentration inequality on the bounds of the variance of gradients. Our proofs of Lemma 1 and Theorem 1 in Appendix E page 18-21 show how the $1/sqrt{n}$ comes up. In assumption (A3), We use $o(\gamma^2)$ instead of $o(\gamma)$ just assume that this variance residual term is smaller than the error rate we need. We verify that this term is extremely small empirically as to $10^{-8}$, therefore the assumption is mild if we assume it is a small order of $o(\gamma^2)$. Therefore this is a mild assumption and doesn’t affect the sample complexity bound rate of $n$.
> >
> > Question 5: Not clear where assumption A3 is used in the proof. Apparently A3 is used in proof of Theorem 3 but A3 is also needed in proof of Theorem 1. It is stated in proof of Theorem 3 that "by Assumption A3.1 we know all input features and weights are bounded." but that is not what A3 states.
> >
> > Answer: Assumption (A3) is used in the derivation of $E(u^2)$ part. In the proof, we first prove a key Lemma 1, which is the keystone of the proof of Theorem 1. And the bottom equations on page 18 shows how we use assumption (A3), which is (A.L.3) for Lemma 1 in the context; In Theorem 3, (A3.1) is the regularity conditions of bounded weights, which we stated in Appendix A.4 page 13.
> >
> > Question 6: The theoretical claim is that l2 norm of weights determines generalization error but experiments use l1 norm. Why?
> >
> > Answer: Thanks for pointing it out. Both $\ell_1$ and $\ell_2$ regularization help control the $\ell_2$ norm of the weights, and they are both effective in the aspect of regularization. We have updated section 5 accordingly.
> >
> > Question 7: Analysis is for regression tasks but experiments are on classification.
> >
> > Answer: Regression tasks and classification tasks share the same model structure except for the output lier. And they Therefore we believe our key observations that attention weights help constrain the magnitude of parameter space also applies to the classification task. Therefore we choose widely-used benchmark datasets to validate our theoretical result.
> >
> > Question 8: Why noisy MNIST ? Why not on the original MNIST ? It is obvious that paying attention on the digit blocks and ignoring the uniform noise blocks will help.
> >
> > Answer: noisyMNIST introduces larger images with noises to the MNIST dataset. Thus for the baseline model, it is more challenging to locate the relevant pixels and predict accordingly. This exactly demonstrates the usefulness of self-attention model in this case. With the transformer-type self-attention model, we can calculate the attention weights precisely, using a smaller parameter size to achieve a more accurate classification.

---

### Official Review · AnonReviewer1 · 2020-10-31
**Very strong assumptions without surprising results**

**Rating:** 5
**Confidence:** 3

**Review:**

This paper provides theoretical analysis of the sample complexity and the loss landscape of attention mechanism. Under several assumptions, the paper claims that attention models have lower sample complexity than non-attention alternatives. Besides, it also shows every local minimum of the attention model has low prediction error. The theoretical findings of the paper can further provide guidelines for designing attention models. A few empirical studies are conducted to confirm the theoretical findings.

The theoretical results rely on a few assumptions, which I am afraid are quite strong.

1. (A1) assumes the data is generated by the attention model plus sub-Gaussian noise, which seems very strong to me. Since your data is obtained from an attention model, it is natural that an attention model can fit the data better with less data than a non-attention alternative. In other words, the sample complexity comparison (Theorem 1 vs Corollary 1) is not fair to me.

2. (A3) looks weird to me, and I don’t see any explanation in the paper. What does this complex assumption mean? Is it just for conquering a step in the proof?

3. Why is the assumption (B1) hidden in the text but not listed in parallel with other assumptions?

4. Again, I have difficulty in understanding the complex (A5).

My feeling is that the author makes strong and hard-to-interpret assumptions, in order to apply the classical learning theory to show the results. The findings are not surprising nor inspiring that the guidance are just common practices in real applications.

When you talk about the loss landscape, is it possible to draw some figures to compare the loss surface of attention models with that of non-attention models? An example would be in this paper: https://papers.nips.cc/paper/7875-visualizing-the-loss-landscape-of-neural-nets.pdf

Typo: The Attention weights are mainly contribute —> contributed.

---

> ### Author Response · Authors · 2020-11-13
> **Detailed response to Reviewer 1 part 1**
>
>
> Thank you very much for your valuable comments.
>
> First, thanks for pointing out your concern about the assumptions. Due to the page limit, detailed justifications on the assumptions are in Appendix A.1 on page 12.
>
> In what follows, we repeat your comments and explain how we addressed them.
>
> Question 1: (A1) assumes the data is generated by the attention model plus sub-Gaussian noise, which seems very strong to me. Since your data is obtained from an attention model, it is natural that an attention model can fit the data better with less data than a non-attention alternative. In other words, the sample complexity comparison (Theorem 1 vs Corollary 1) is not fair to me.
>
> Answer: (A1) assumes the attention model is the generative model. First, there are many application cases for this model. For example, in our experiments on noisyMNIST dataset, when we don’t know where the digit is located. In this case, the baseline model has the same expressiveness power on this dataset, and our theory explains why such design of self-attention model learning the concentration of all images helps us train a better model. Second, we think generative assumption (A1) is not a strong assumption, considering the massive literature proves the strong expressiveness of neural nets. Specifically for two-layer neural nets and transformer model, Barron & Klusowski 2018 shows that general bounded functions with a Fourier representation on [-1,1] can be well approximated by the defined two-layer network. For the transformer model, Yun et al.(2019) prove the strong expressiveness of transformer type self-attention model for any bounded sequence-to-sequence functions.
>
> Question 2: (A3) looks weird to me, and I don’t see any explanation in the paper. What does this complex assumption mean? Is it just for conquering a step in the proof?
>
>
> Answer: (A3) is explained and justified in detail in the Appendix. The high-level idea of the assumption is that the hidden layers are sufficiently overparameterized such that the linear combinations of them can explain most of the true hidden units. We justify this assumption both theoretically and empirically. Theoretically, we show that when $d$ is large, given the $2^d$ linear regions and strong expressiveness of $\{\phi_{1\dots d}(\langle w^{(1)},x\odot f(x)\rangle)\}$, the residual vector is a $o(\gamma)$ term with respect to all directions in $\mathbb{R}^d$. Also, it coincides with the assumption in Nguyen et al. 2017 and the theoretical result in Allen-Zhu et al. 2019. Empirically, we also evaluate the residual term in a two-layer network with self-attention model. It shows the largest eigenvalue of the residual covariance matrix is as small as $3.13*10^{-8}$. Therefore we can assume it to be very small.

---

> > ### Author Response · Authors · 2020-11-13
> > **Detailed response to Reviewer 1 part 2**
> >
> >
> > Detailed justification is provided on Page 12 Appendix A.1. We also attached it here.
> >
> >
> > (A3) assumes that $\{\phi_{1\dots d}(\langle w^{(1)}, x_i\odot f(x_i)\rangle)\}$ obtain sufficient expressiveness to predict the output $y$ when sample size is sufficiently large. %We should point out that $\gamma$ in (A2) can be arbitrarily small when the network is overparameterized.
> >
> > In an overparameterized network with large $d$, we know $\{\phi_{1\dots d}(\langle w^{(1)}, x_i\odot f(x_i) \rangle)\}$ have up to $2^d$ number of linear regions over $n$ samples. And (A2) assumes these linear regions space all directions of $\phi(\langle w^{(1)\star}, x_i\odot f(x_i)\rangle)$ up to a $o(\gamma)$ term. It says that given the $2^d$ linear regions and strong expressiveness of $\{\phi_{1\dots d}(\langle w^{(1)},x\odot f(x)\rangle)\}$, the residual vector is a $o(\gamma)$ term with respect to all directions in $\mathbb{R}^d$. This assumption is parallel to the full column rank condition in Nguyen et. al. 2017, where we essentially assume in overparameterized network, the linear combination spans all directions in $\mathbb{R}^p$. Allen-Zhu et. al. 2019 also shows the fact in their Lemma B.1 and Corollary B.2 that in overparameterized networks, in every small region of parameter space, there exists set of parameters with good prediction. And it leads to a good landscape in each neighborhood. These results all indicate that it is reasonable to assume sufficient expressiveness of $\phi(\langle w^{(1)}, x \odot
> > a \rangle)$ in overparameterized networks as $n \to \infty$. What's more, with required large sample size, we can also straightforwardly evaluate this assumption by checking whether these $\phi(\langle w^{(1)},x \odot a \rangle)$ spread out the whole space and their linear combinations have good estimation on $y$. All these results guarantee that (A3) can be achieved in overparameterized networks. We also empirically validate (A3) by computing the largest eigenvalue of such conditional covariance in (A3). We generate a two-layer network under self-attention model (4) with $n=500$, $d=256$, $p=2142$, $d_v=100$. We choose the set of parameter same with our experiment setup in Section 6, and hidden layers are sufficient overparameterized with $p=2142$. Then we compute the empirical conditional covariance $\hat{\lambda}_{\max}(Var(\phi(\langle w^{(1)\star}, x_i\odot f(x)\rangle)|\phi(\langle w^{(1)}, x\odot f(x)\rangle))=3.13*10^{-8}$. This result indicates that the largest eigenvalue term $\lambda_{\max}(Var(\phi(\langle w^{(1)\star}, x_i\odot f(x)\rangle)|\phi(\langle w^{(1)}, x\odot f(x)\rangle))$ is small enough to be assumes as $o(\gamma^2)$ as in (A3).
> >
> >
> > Question 3: Why is the assumption (B1) hidden in the text but not listed in parallel with other assumptions?
> >
> >
> > Answer: The rigorous version of Assumption (B1) is deferred to Appendix A.2. page 13. Please check (A1.1)-(A1.3) for a complete version of assumption (B1). They are mild assumptions on network weights and the distribution of features. We summarize them in words in the main paper to improve the readability of the paper.
> >
> >
> > Question 4: Again, I have difficulty in understanding the complex (A5).
> >
> >
> > Answer: (A5) is parallel to (A3), as we justified above.
> >
> >
> > Question 5: My feeling is that the author makes strong and hard-to-interpret assumptions, in order to apply the classical learning theory to show the results. The findings are not surprising nor inspiring that the guidance are just common practices in real applications.
> >
> >
> > Answer: Finally, we would like to further discuss the contribution of the paper. Our results on useful in different aspects: (1) Theoretically we observe a key fact why attention model achieves a smaller complexity: The attention weights help constrain the magnitude of the parameter space, thus control the variance of the gradient. This fact leads to a faster concentration on the gradient, and eventually, a small sample complexity bound. (2) Our analysis emphasizes the importance of concentration in attention weights in model design. In practice, the temperature in softmax matrix or sparse attention design achieves such concentration effects. Our result provides theoretical and empirical justification for such design.
> >
> >
> > Question 6: When you talk about the loss landscape, is it possible to draw some figures to compare the loss surface of attention models with that of non-attention models? An example would be in this paper: https://papers.nips.cc/paper/7875-visualizing-the-loss-landscape-of-neural-nets.pdf.
> >
> >
> > Answer: Thanks for pointing out the visualization of the loss landscape. We will further investigate the visualization in the future version.

---

### Decision · Program_Chairs · 2021-01-07
**Final Decision**

**Decision:**

Reject

**Comment:**

The three reviewers seem to reach a consensus that the assumptions made in the paper are too strong and hard to interpret. In particular, R1&R2 made the comments that the generative model for the data by itself uses attention, which seems to be make the comparison unfair. The authors seem to argue that the attention model is still expressive enough, which in the AC's opinion could be true but does not justify the use of the generative model when comparing the sample complexity of the two methods. The reviewers also pointed out a few other limitations of the paper. The AC mostly agrees with the reviewers' points (though perhaps with one or two exceptions.) In summary, I think the assumption of using an attention model itself seems to be a big enough issue that makes the paper not read for publication at ICLR.